# Perceived facilitators and barriers among physical therapists and orthopedic surgeons to pre-operative home-based exercise with *one* exercise-only in patients eligible for knee replacement: A qualitative interview study nested in the QUADX-1 trial

**Rasmus Skov Husted**[1,2,3]*, **Thomas Bandholm**[1,2], **Michael Skovdal Rathleff**[4,5,6], **Anders Troelsen**[3], **Jeanette Kirk**[1,2]

**1** Department of Clinical Research, Copenhagen University Hospital Amager-Hvidovre, Hvidovre, Denmark, **2** Department of Physical and Occupational Therapy, Physical Medicine & Rehabilitation Research-Copenhagen (PMR-C), Copenhagen University Hospital Amager-Hvidovre, Hvidovre, Denmark, **3** Department of Orthopedic Surgery, Clinical Orthopedic Research Hvidovre (CORH), Copenhagen University Hospital Amager-Hvidovre, Hvidovre, Denmark, **4** Research Unit for General Practice in Aalborg, Department of Clinical Medicine, Aalborg University, Aalborg, Denmark, **5** Department of Occupational Therapy and Physiotherapy, Aalborg University Hospital, Aalborg, Denmark, **6** Department of Health Science and Technology, Aalborg University, Denmark

* rasmus.skov.husted@regionh.dk

## Abstract

### Aim

Clinical guidelines recommend non-surgical treatment before surgery is considered in patients eligible for knee replacement. Surgical treatment is provided by orthopedic surgeons and exercise therapy is provided by physical therapists. The aim of this study was to identify perceived facilitators and barriers–among orthopedic surgeons and physical therapists–towards coordinated non-surgical and surgical treatment of patients eligible for knee replacement using pre-operative home-based exercise therapy with *one* exercise.

### Methods

This qualitative study is embedded within the QUADX-1 randomized trial that investigates a model of coordinated non-surgical and surgical treatment for patients eligible for knee replacement. Physical therapists and orthopedic surgeons working with patients with knee osteoarthritis in their daily clinical work were interviewed (one focus group and four single interviews) to explore their perceived facilitators and barriers related to pre-operative home-based exercise therapy with *one* exercise-only in patients eligible for knee replacement. Interviews were analyzed using thematic analysis.

**Data Availability Statement:** All relevant data are within the manuscript and its Supporting Information files.

**Funding:** This work was supported by The Capital Region's strategic funds, https://www.regionh.dk/til-fagfolk/Forskning-og-innovation/finansiering-og-fonde/s%C3%B8g-regionale-midler/Sider/Region-Hovedstadens-forskningsmidler.aspx; grant number: [R142-A5363] (TB); and The Capital Region's fund for cross-continuum research, https://www.regionh.dk/tfe/tvaerspuljen/Sider/default.aspx; grant numbers: [P-2018-1-02, P-2019-1-03] (TB). The funders had no role in study design, data collection and analysis, decision to publish, or preparation of the manuscript.

**Competing interests:** Dr. Troelsen reports personal fees from Zimmer Biomet, grants from Zimmer Biomet, personal fees from European Knee Society, outside the submitted work. This does not alter our adherence to PLOS ONE policies on sharing data and materials.

## Results

From the thematic analysis three main themes emerged: 1) *Physical therapists' dilemma with* <u>one</u> *home-based exercise*, 2) *Orthopedic surgeons' dilemma with exercise*, and 3) *Coordinated non-surgical and surgical care*.

## Conclusion

We found that the pre-operative exercise intervention created ambivalence in the professional role of both the physical therapists and orthopedic surgeons. The physical therapists were skeptical towards over-simplified exercise therapy. The orthopedic surgeons were skeptical towards the potential lack of (long-term) effect of exercise therapy in patients eligible for knee replacement. The consequence of these barriers and ambivalence in the professional role is important to consider when planning implementation of the model of coordinated non-surgical and surgical treatment.

## Trial registration

ClinicalTrials.gov, ID: NCT02931058.

## Introduction

Knee osteoarthritis (OA) is a growing challenge for the health care system and more knee replacements (KR) are performed each year to treat the condition with an estimated increase of 69% from 2012 to 2050 in the United States [1,2]. A key feature of knee OA is knee pain which is often associated with decreased quality of life, physical activity and muscle strength. Due to pain and physical impairments it may increase the risk of sick leave and early retirement [3,4]. Traditionally, patients eligible for KR are provided highly specialized surgical treatment to help overcome their knee OA-related pain and concomitant symptoms [5,6]. Patients, not eligible for KR, can be referred to non-surgical treatments such as exercise therapy and weight loss [7]. Both international and national guidelines recommend non-surgical treatment (i.e. exercise therapy, weight loss, self-management and education) before surgery is considered in patients eligible for KR [7–12]–and recent studies show that exercise therapy can provide clinically relevant improvements in knee OA symptoms in patients eligible for KR [13–18]. Despite this, it is estimated that up to 25% of patients could be inappropriately receiving KR prior to the recommended non-surgical treatment [19]. Eligibility for KR is based on several factors including; the patient's medical history (i.a. knee pain, quality of life, limitations in daily living), physical examination (i.a. active and passive range of motion, palpation), effect of previous non-surgical treatment [7,8,10] and x-rays [20,21]. Knee OA is roughly categorized as mild-to-moderate or severe and the main difference between the two categories is the level of symptomatology (e.g. knee pain levels) [22]. Patients with severe knee OA are more likely to be deemed eligible for KR and undergo surgery [23–25]. Most patients referred to KR present with severe radiographic knee OA (i.e. clear joint space narrowing, osteophytes, sclerosis and joint deformity) as this is believed to be a good indication as to whether KR will be an effective treatment [26].

Improving the coordination of treatment across health care sectors is vital in order to improve outcomes for patients with knee OA [27]–and it can be achieved. For example, rehabilitation exercise therapy after KR is coordinated across health care sectors in current clinical

practice. This is not the case for the treatment of patients with knee OA who are potential candidates for KR. Optimally, orthopedic surgeons should refer patients potentially eligible for KR to initial non-surgical treatment (e.g. exercise therapy) in their municipality—according to guideline recommendations [7,8] and then re-evaluate the need for surgical treatment by shared decision making, based on changes in symptoms [21]. In order to organize a coordinated care pathway like this, stakeholder involvement is fundamental [28]. Involving stakeholders will help identify and manage facilitators and barriers related to the coordinated care pathway under study [29]. In the care of patients with knee OA, the stakeholders in daily clinical work are the orthopedic surgeons and physical therapists. Thus, their views and thoughts on a new care initiative are highly important if this one initiative is to be effectively implemented and adopted in clinical practice [30].

Previous reports investigating facilitators and barriers among orthopedic surgeons suggest a number of challenges in the use of non-surgical treatment in patients with knee OA [27,31]. For example, "No effect of physical therapy when there is an obvious loss of cartilage" and "Lack of visibility into physical therapies" were associated with decline in referrals to physical therapy and reported as a barriers [27]. In the way that surgery is considered important for the profession of orthopedic surgery [5], exercise therapy is considered important for the profession of physical therapy [32,33]. However, a study in patients with shoulder pain, suggests that physical therapists have barriers concerning simplified exercise interventions [32], that is, interventions with a minimal number of exercises and limited consultation time between the physical therapist and patient. The above implies that orthopedic surgeons and physical therapists may have barriers concerning coordinated non-surgical care in the form of simplified pre-operative exercise therapy in patients eligible for KR. Context-specific screening for facilitators and barriers is important to help facilitate implementation in clinical practice and optimize coordination of treatment [34,35].

### Aim

The aim of this study was to identify perceived facilitators and barriers–among orthopedic surgeons and physical therapists–towards coordinated non-surgical and surgical treatment of patients eligible for knee replacement using pre-operative home-based exercise therapy with *one* exercise.

## Methods

### Context: The QUADX-1 trial

This qualitative study is an embedded part of a "parent" randomized trial (the QUADX-1 trial) investigating the dose-response relationship of pre-operative exercise therapy prior to potential KR in patients with severe knee OA [36]. The trial employs a model of coordinated non-surgical and surgical treatment where orthopedic surgeons re-evaluate patients' need for surgery following exercise in the municipality. In the Danish health care system, non-surgical treatment is performed in the municipalities and surgical treatment is performed at the hospitals. To ensure coherent care pathways with high quality for patients, cross-sector coordination of treatment is essential. In the QUADX-1 trial, the patients exercise unsupervised at home for twelve weeks after an initial exercise instruction by a physical therapist. At four and eight weeks, the patients have follow-up consultations with the physical therapist. The project is designed as an intervention trial with concurrent identification of perceived facilitators and barriers (the present qualitative study), also referred to as a hybrid I design [37]. The exercise intervention consists of *one* muscle strengthening exercise–seated knee-extensions using an elastic exercise band as resistance. This *one* specific exercise was chosen based on the concept

of "less is more", as it was considered pragmatic and simple. That is, it is easy to set up at home, easy to remember how to perform, requires little intellectual effort and is easy to master [38]. Further, an exercise intervention comprising *one* exercise was chosen as compliance to home-based exercise is reported to be poor [11,39–41] and an intervention of one exercise could increase adherence. Additional details on the QUADX-1 trial and the *one* knee-extension exercise is available in the published open access trial protocol [36].

## Design

This qualitative study involves the analysis of data collected during interviews with the two main stakeholders: orthopedic surgeons and physical therapists. Interview is a recognized qualitative method for obtaining in-depth knowledge about stakeholder's feelings, experiences and attitudes [42,43]. We performed interviews to understand the perceived barriers and facilitators among the orthopedic surgeons and physical therapists regarding this novel coordination of surgical and non-surgical treatment prior to the beginning of the parent trial. By using interviews RSH was able to interpret words as well as body language to obtain a better understanding. Further, it provided an opportunity to create trust between RSH and the participants which is important when talking about facilitators and barriers related to work, habits and potential associated feelings. This is an acknowledged design when investigating facilitators and barriers in an implementation context [37]. The qualitative study is reported according to the Standards for Reporting Qualitative Research: A Synthesis of Recommendations (SRQR) checklist [44] (S1 File).

## Study setting

The study was carried out in Denmark, where the health care system is publicly funded from taxes, enabling the Danish welfare state to provide free treatment for all citizens. The orthopedic department where this study was performed is an integrated part of the hospital and has more than 45,000 ambulatory visits and around 7,000 operations are performed every year. All municipalities have rehabilitation centers where patients are referred to outpatient rehabilitation subsequent to treatment at the hospital, for example KR.

## Recruitment and study participants

We recruited six physical therapists from three municipalities in the capital region of Denmark and four orthopedic surgeons from one university hospital in the capital region of Denmark involved in the QUADX-1 trial [36] (Table 1). Inclusion criteria: participants had to be involved in the QUADX-1 trial, as the intervention under study was not implemented in routine clinical practice. No exclusion criteria were applied. Thus, the six physical therapists and four orthopedic surgeons represents all possible participants as we sought situational representativeness rather than demographic representativeness [45]. We recognize that a sample size of ten participants could be a limitation and inadequate to certify data saturation. However, we consider this necessary as this was the only possibility to investigate facilitators and barriers among the orthopedic surgeons and physical therapists involved in the QUADX-1 trial. When interpreting the results this should be kept in mind. Participants were contacted by the primary investigator and interview moderator (RSH) by e-mail with an invitation to participate in the interviews between November 2016 and June 2017. RSH sent the invitations because he would be conducting the interviews. All invited participants accepted. All eligible physical therapists had daily clinical work with patients diagnosed with knee OA and rehabilitation following KR. All eligible orthopedic surgeons had daily clinical work with patients potentially eligible for KR due to knee OA symptoms. A random sample of patients participating in the

**Table 1. Participant demographics.**

| Participants | Sex | Age (years) | Clinical experience (years) | Experience as knee replacement specialist (years) |
|---|---|---|---|---|
| Physical therapist 1 | F | 44 | 17 | Na. |
| Physical therapist 2 | F | 31 | 6 | Na. |
| Physical therapist 3 | F | 38 | 11 | Na. |
| Physical therapist 4 | M | 46 | 19 | Na. |
| Physical therapist 5 | M | 30 | 1 | Na. |
| Physical therapist 6 | M | 32 | 1.5 | Na. |
| Orthopedic surgeon 1 | M | 35 | 3 | 1 |
| Orthopedic surgeon 2 | M | 59 | 30 | 19 |
| Orthopedic surgeon 3 | M | 55 | 27 | 19 |
| Orthopedic surgeon 4 | M | 38 | 6 | 1 |

F = female, M = male, Na. = not applicable.

QUADX-1 trial were also interviewed about their perceptions of facilitators and barriers towards coordinated non-surgical and surgical treatment using pre-operative home-based exercise therapy with *one* exercise. This work is as yet unpublished.

### Interviews: Focus group and single interviews

We aimed to use focus group interviews for all participants as the purpose of the interviews was to explore the perceived facilitators and barriers, associated feelings, opinions and attitudes of the health care professionals on the coordinated non-surgical and surgical treatment investigated in the QUADX-1 trial. Focus group methodology is considered an appropriate method for this purpose because participants can freely express and discuss their experiences as well as listen to the experiences of the other participants. It is therefore particularly suitable to collect data on social groups, interactions, interpretations and norms [42,43,46]. We completed the focus group interview with the physical therapists as planned, but it proved practically impossible with the orthopedic surgeons due to very tight work schedules. As a compromise, we conducted single interviews instead because this method is suitable for producing in-depth data on a particular phenomenon or topic [47]. Both the focus group interview and the single interviews were guided by semi-structured interview guides with open-ended questions (S2 and S3 Files). The interview guides were based on literature related to the organization and recommended order of non-surgical and surgical treatment in patients eligible for knee replacement [5–9], effectiveness of exercise therapy in patients with knee OA [13–16], traditional organization of exercise therapy [33,48,49], knee replacement surgery [5,6,19], previous identified barriers and facilitators towards exercise therapy and surgery [27,31,32] and shared decision-making [21] in patients eligible for knee replacement. A "funnel approach" was used at all interviews starting with broad open-ended questions followed by probing and sensitizing questions aiming to elicit deeper and more detailed information [42].

### Procedures

The interviews took place before the first patient was enrolled in the QUADX-1 trial to ensure no experience with the trial among the participants. Thus, the interviews only relate to their preconceptions and not later experiences during the trial. The interviews took place in meeting rooms at a university hospital in the capital region of Denmark. The two interview guides were developed in an iterative process by RSH, TB and JK informed by literature and clinical experience to steer the interviews towards the phenomena of interest. Before the interviews, the two

interview guides were piloted by RSH and JK in two single interviews with health care professionals to revise poorly formulated questions after which they were re-piloted. The focus group interview lasted two hours (including two breaks) and the single interviews lasted between 30 and 40 minutes. At the focus group interview, the moderator (RSH) facilitated the dialogue while JK observed the interview and took notes of topics important to pursue. RSH and JK went through these notes in the two breaks in the focus group interview and adjusted the interview to ensure that these topics were included (e.g. topics not pursued by RSH due to preconception as a physical therapist). The single interviews were conducted by RSH and at the first single interview, JK observed and took notes of important topics, which were used to qualify the following interviews. Following every interview, RSH and JK listened to the audio file and adjusted the interview guide based on new important topics (S4 File). The interviews were recorded with a digital voice-recorder (Philips Voice Tracer LFH0882) and afterwards transcribed verbatim by RSH.

## Data analysis

Fully transcribed interviews were brought together into one text constituting the unit of analysis. Before analyzing the interviews, RSH read the data material through several times to obtain a sense of the whole. The transcribed interviews were analyzed by RSH and JK using inductive thematic analysis to group the data into sub-themes and themes [50]. The analytical process involved 1) dividing the text into meaning units, 2) condensing meaning units, 3) abstracting and coding the condensed meaning units, 4) sorting codes based on similarities and differences, 5) sorting codes into sub-themes and themes. Tentative sub-themes/themes were discussed by RSH, TB and JK through a process of reflection and discussion. These discussions facilitated an iterative process in which RSH and JK re-analyzed meaning units and codes and re-coded the data based inputs from the discussions [51]. Themes were then revised and agreed on to strengthen the validity of the results. 6) Finally, the latent content (underlying meaning) of the sub-themes was formulated into themes. The final themes were discussed and agreed upon by all members of the research team (Table 2). Through this process, it was possible for RSH to put his preconception in dialogue with the text (fusion of horizons). Thus, the understanding of physical therapists and orthopedic surgeons perceived facilitators and barriers towards coordination of surgical and non-surgical treatment gradually changed [52]. As an example, one preconception was related to orthopedic surgeons' view on exercise as not being

**Table 2. Example of inductive thematic analysis.**

| Meaning unit | Condensed meaning unit: description close to the text | Condensed meaning unit: interpretation of the underlying meaning | Sub-theme | Theme |
|---|---|---|---|---|
| Physical therapist 2, focus group interview: "*Well, I have been thinking. They (the patients) come for instruction in only one exercise and we are not supposed to consider all the other things. Eh, all the questions they might have regarding other painful areas of their body, whatever the reason. I have certainly been thinking that I wanted to examine them more closely in general and in relation to their knee OA. Yes, now it's only this one exercise they get.*" | The physical therapists are only supposed to give instructions in one exercise. Not consider other questions or disorders the patients might have. | The physical therapists want to examine the patients for other disorders and not only provide instruction for one exercise. | Professional role as a physical therapist is simplified. | Physical therapists' ambivalence in their professional role. |

Example of inductive thematic analysis to group data into sub-themes and themes as described by Nowell et al. [50].

useful in patients with severe knee OA. This preconception changed when the effect of exercise or lack hereof was mentioned as useful in the decision on surgery or not. An audit trail of the thematic analysis and a section on trustworthiness is provided as supporting information (S5 File).

## Ethics

The study was performed according to the Helsinki Declaration [53]. Before undertaking the interviews, all participants were provided oral and written information about the aim of the study, procedures to be undertaken, potential risks and benefits of participation, expected duration of the study and extent of confidentiality of personal identification, assured that participation was voluntary and that they could withdraw consent at any time during the interview. All invited participants were allowed a minimum of 24 hours to consider participation. All participants were informed about anonymity and confidentiality and gave written informed consent to participate in the interviews. All participants are pseudo-anonymized and reported data are de-identified (no mentioning of names or date of birth). Data were stored on a file drive secured by log-in. The study has been approved by The National Committee on Health Research Ethics (Protocol no.: H-16025136).

## Results

The thematic analysis showed three main themes with nine associated sub-themes: Theme 1) *Physical therapists' dilemma with one home-based exercise;* Sub-themes 1) *Supporting patient self-management is a physical therapy core skill*, 2) *Professional role as a physical therapist is simplified*, 3) *Skepticism towards one home-based exercise* and *4) Patient preferences*. Theme 2) *Orthopedic surgeons' dilemma with exercise*; Sub-themes 5) *Skepticism towards (long-term) effect of exercise in patients with severe knee OA*, 6) *Patients motivation* and 7) *Different purposes of referring a patient to exercise*. Theme 3) *Coordinated non-surgical and surgical care*; Sub-themes 8) *Orthopedic surgeons' skepticism to the content of the exercise treatment they refer to* and 9) *Responsibilities in coordinated care and engagement in the care pathway* (Table 3). The themes and sub-themes represent different perceived facilitators and barriers among orthopedic surgeons and physical therapists towards home-based pre-operative exercise in patients eligible for KR.

**Table 3. Themes and associated sub-themes.**

| No. | Themes | No. | Sub-themes |
|---|---|---|---|
| 1 | Physical therapists' dilemma with *one* home-based exercise | 1 | Supporting patient self-management is a physical therapy core skill |
| | | 2 | Professional role as a physical therapist is simplified |
| | | 3 | Skepticism towards *one* home-based exercise |
| | | 4 | Patient preferences |
| 2 | Orthopedic surgeons' dilemma with exercise | 5 | Skepticism towards (long-term) effect of exercise in patients with severe knee OA |
| | | 6 | Patients motivation |
| | | 7 | Different purposes of referring a patient to exercise |
| 3 | Coordinated non-surgical and surgical care | 8 | Orthopedic surgeons' skepticism to the content of the exercise treatment they refer to |
| | | 9 | Responsibilities in coordinated care and engagement in the care pathway |

Themes and associated sub-themes from the inductive thematic analysis.

Among the physical therapists' and orthopedic surgeons' different facilitators and barriers relate to how coordinated non-surgical and surgical care with home-based exercise therapy with *one* exercise creates a dilemma leading to ambivalence in their professional roles. Ambivalence is defined as a condition that occurs when you have two conflicting feelings or attitudes at the same time. With ambivalence you will have difficulty making decisions, as all solutions seem equally good or equally bad [54].

### Theme 1 –Physical therapists' dilemma with *one* home-based exercise

**Sub-theme 1: Supporting patient self-management is a physical therapy core skill.** One sub-theme that emerged from the focus group interview was that the physical therapists are conscious about the importance of educating and providing patients with tools to self-manage their condition. The following excerpt illustrates the physical therapists' attitude towards this:

"*We are focused on this (self-management) right from the beginning. At the same time, we tell them (the patients) that rehabilitation is not finished after an interim exercise program in the municipality and that it is necessary to continue exercising to get the full benefit.*" (Physical therapist 2, focus group interview).

The physical therapists mention the importance of ensuring patient adherence to exercise and teaching patients how to adjust their treatment (exercise) properly in line with their symptoms. Different pedagogic approaches were discussed between the physical therapists to achieve this. Typically, this is a process going from a lot of supervision and guidance towards less and less as the patients become independent. One physical therapist expresses:

"*I think that one of my greatest tasks, together with the patient, is to make the patient independent so that they are able to manage without me when they have finished supervised treatment. . . I think this one of the most important tasks. It is something we focus on right from the moment they come through the door.*" (Physical therapist 1, focus group interview).

As the physical therapists are aware of this, they embrace this skill and express that it is important to give patients a sense of responsibility for their own treatment and to teach them principles of self-management of their condition. In this way, even though they distance themselves from the patients, they keep some control over the patient's treatment, and it becomes a potential facilitator.

**Sub-theme 2: Professional role as a physical therapist is simplified.** In the present trial, the diagnosis and treatment are already defined, meaning that the physical therapists do not need to clinically examine the patients. They can go right ahead and instruct the patients on how to do the *one* exercise at home. The role of the physical therapist becomes (somewhat) predefined and pedagogical compared to more traditional clinical practice with frequent adjustments to treatment. This was mentioned as unusual practice by the physical therapists:

"*Well, this limitation I have been given and the desire to carry out an examination, and other things which you now have to avoid, is an unusual role that you have to get used to.*" (Physical therapist 3, focus group interview).

The limited face-to-face contact between patient and physical therapist where they only see each other three times over the course of three months, gives rise to several concerns among the physical therapists. They express concern about the quality (and thus effectiveness) of the exercise (treatment) when it is primarily home-based as they are not there to ensure high

quality in the exercises and provide timely adjustments. Related to this is a concern about the limited number of predefined consultations with the possibility of supervising and adjusting the exercise. A physical therapist explained:

> *"Well, I have been thinking. They (the patients) come for instruction in only one exercise and we are not supposed to consider all the other things. All the questions they might have regarding other painful areas of their body, whatever the reason. I have certainly been thinking that I wanted to examine them more closely in general and in relation to their knee OA. Yes, now it's only this one exercise they get."* (Physical therapist 2, focus group interview).

The predefined and advisory role with a limited number of consultations challenges and simplifies their professional role and, thus, becomes a potential barrier.

**Sub-theme 3: Skepticism towards one home-based exercise.**   The physical therapists express concern regarding exercise treatment with *one* exercise as they think it is a rigid treatment limiting usage of their professional skills. One physical therapist expressed:

> *"Well, everything depends on this <u>one</u> exercise you know."* (Physical therapist 6, focus group interview).

The physical therapists discussed the possibility of changing the exercise (e.g. if performed with low quality) or add more exercises based on the patient's symptom response and preferences. The physical therapists explained that this is how "normal" clinical practice is carried out and an integrated part of the physical therapists' professional work. This was shown in statements such as this:

> "*Well, often something (an exercise) is effective, but if you find out that it is not, then you know that you have other exercises which would be effective, and you might well add these to the exercise program.*" (Physical therapist 6, focus group interview).

A home-based single exercise intervention without the option of exercise adjustment or the addition of other exercises becomes a potential barrier to physical therapists.

**Sub-themes 4: Patient preferences.**   Despite the above-mentioned barriers regarding a simplified professional role and *one* home-based exercise, there was broad consensus among the physical therapists that it is good to have two treatment options (supervised and home-based). A physical therapist said:

> "*Yes, and it is not as if you can say it is good for everyone. It is good for some yes. And it is good for those who don't want to be absent from work or who would like to stay in their holiday house, and therefore cannot come to the gym. And where we can also see that the exercise is done satisfactorily. But it is definitely not for everyone*". (Physical therapist 1, focus group interview).

Further, the physical therapists also discussed other facilitators related to the *one* home-based exercise approach. Providing patients with *one* exercise keeps the intellectual effort required to a minimum while also taking less time to complete. The physical therapists describe the exercise as tangible, which could increase patient ownership of the exercise, potentially leading to improved exercise adherence and thereby, treatment effect. One physical therapist stated:

> "*I think it is a very tangible exercise they (the patients) have to go home and do, and it is also easy. And I think this must be good for the patient. . .. That there are not so many questions*

*when they get home. How was I was supposed to do the exercise?. . . I think this actually is a strength with the exercise.*" (Physical therapist 1, focus group interview).

In continuation of this, the physical therapists also mention that the *one* exercise could be very important:

"*We only give them <u>one</u> exercise; they don't have anything else. I think this is the reason they get it done. . . Also, the fact that they have <u>one</u> exercise. This makes it very important.*" (Physical therapist 3, focus group interview).

This underlines that according to the physical therapists not all patients are candidates for home-based exercise, which also supports the option of a stratified treatment approach (two treatment options). The possibility of two treatment options and thus a better chance of providing treatment suiting individual patient preferences becomes a potential facilitator.

In summary, a single exercise home-based intervention creates a dilemma among the physical therapists. On the one hand the physical therapists perceive *the importance of providing patients with tools for self-management*, *the advantage of having two treatment options to meet patient preferences* and *the potential advantages of providing patients with only <u>one</u> exercise* as facilitators for implementing the one exercise. These factors support the simplified treatment approach among the physical therapists and their view on their professional role. On the other hand, the physical therapists believe that the simplified treatment approach *simplifies their professional role*, *limits contact time with patients* and *providing only <u>one</u> exercise limits use of professional skills*. These barriers challenge the physical therapists creating ambivalence in their professional role.

## Theme 2: Orthopedic surgeons' dilemma with exercise

All the orthopedic surgeons are aware that clinical guidelines recommend non-surgical treatment (e.g. exercise therapy and weight loss) in all patients before surgical treatment is considered. An orthopedic surgeon explains:

"*. . .the guidelines state that the patient must be offered conservative treatment before surgery.*" (Orthopedic surgeon 1, single interview).

However, the decision of treatment is more complex than simply referring all patients without preceding non-surgical treatment to e.g. exercise therapy. The clinical experience and expertise of the orthopedic surgeons and patient preferences also affect the treatment decision.

**Sub-theme 5: Skepticism towards (long-term) effect of exercise in patients with severe knee OA.**   Among the interviewed orthopedic surgeons there is variation in the views and opinions regarding exercise and the applicability of this in patients with severe knee OA. In general, the orthopedic surgeons consider exercise to be an integrated part of the treatment options they use in patients with mild-to-moderate knee OA. An orthopedic surgeon explained:

"*Patients with knee pain, mild-to-moderate osteoarthritis, who haven't tried conservative treatment (non-surgical) are eligible for exercise.*" (Orthopedic surgeon 3, single interview).

If symptoms progress, surgical treatment might be needed. Generally, when assessing the indication for surgery, the orthopedic surgeons agree that knowledge about the effect of exercise treatment provides clinically relevant information aiding the decision to proceed with

surgery or not. One situation discussed by the orthopedic surgeons was that if exercise treatment improves symptoms, then surgery might not (yet) be warranted. Contrary, lack of treatment effect from exercise could strengthen the argument for surgical treatment. One orthopedic surgeon stated:

> "*I can certainly tell you this. If the patient's symptoms are unchanged and they are still in so much pain that the indication for surgery is there, then there is no doubt that surgery is the right treatment. If they have had no effect from exercise.*" (Orthopedic surgeon 3, single interview).

However, there is considerable skepticism among orthopedic surgeons towards exercise as a treatment modality for patients with severe knee OA. The main skepticism is related to a lack of belief in the effectiveness of this treatment. One orthopedic surgeon stated:

> "*Well, no, I am a little in doubt of how effective it (exercise) is. Patients on painkillers with a lot of pain who clearly have arthritis and that type of thing, I don't really believe that any of them can avoid surgery.*" (Orthopedic surgeon 2, single interview).

This skepticism is greater when it comes to long-term effects of exercise. The orthopedic surgeons know that surgery is very effective (when successful) in the long-term. On the contrary, exercise, if effective at all, is only effective when maintained. The following excerpt illustrates the orthopedic surgeon's attitude towards this:

> "*I question whether it is at all fair (to refer patients with severe knee OA to exercise). I think of this when they (the patients) have a lot of discomfort and pain and osteoarthritis and you know it helps a lot of patients with knee arthroplasty. I also think it might well be that exercise helps, but I seriously doubt that the effect will be long-term. I don't have anything to base this on, but . . ..*" (Orthopedic surgeon 2, single interview).

In line with the above and despite the guideline recommendations, it is the general opinion among the interviewed orthopedic surgeons that some patients should not try out exercise treatment but should be offered surgery right away because of the severity of their symptoms, associated x-ray and lack of effect of e.g. analgesic treatment. The orthopedic surgeons state that it would be a waste of resources to do anything else than offer surgery, because they believe that no other treatment is effective enough in relieving the patient's symptoms. If a patient with severe symptoms is referred to exercise treatment, the orthopedic surgeons are convinced that the patient will come back without feeling better. One orthopedic surgeon gave an example:

> "*If, you have a patient, let's say, an 80-year-old male with a really bad varus-knee with bone-on-bone in all three compartments and receiving strong pain medication. Then sometimes you might think, "it is really stupid to refer him to exercise for three months" because it would be a waste of time.*" (Orthopedic surgeon 1, single interview).

Referring patients with severe knee OA to exercise challenges the orthopedic surgeon's professional role and self-image, as it becomes ambivalent to refer patients to exercise, as the guidelines recommend, when they doubt the effectiveness of the treatment. This was evidenced in statements like the one below:

"*Well, for me as a professional, we must at least believe a little in its (exercise) effectiveness before we refer patients to it, when we have the other alternative (surgery).*" (Orthopedic surgeon 2, single interview).

Lack of belief in the effectiveness of exercise for patients with severe knee OA, doubt about the long-term effects of exercise and knowledge about the effectiveness of surgery create skepticism in the orthopedic surgeons, and these become potential barriers to referring patients with severe knee OA to exercise.

**Sub-theme 6: Patients motivation.** A number of points made by the orthopedic surgeons demonstrate the complexity of referring patients to exercise therapy. Patient characteristics can also play a role. The orthopedic surgeons expressed concern in referring older patients with no history of exercising to an exercise intervention, as they were unsure if the patients would adhere or be motivated. This was shown in statements like the following:

"*Well, some of them are not used to exercise. There are a lot of patients in this age-group who are not used to exercise. They have never done it. Nowadays plenty of people exercise, also in their 40's and 50's. But in that generation now in their 60's, many of them have never exercised. I think this makes it more difficult for them, mentally.*" (Orthopedic surgeon 3, single interview).

The orthopedic surgeons also expressed concern for patients still active in the labor market who do not want a prolonged treatment (e.g. because of pressure from their employer or the risk of losing their job). The orthopedic surgeons could be inclined to suggest surgery as they see this as the treatment with the most certain course (time, effect, risks). On the contrary, the effect of exercise is smaller than that of surgery in most cases and the possibility of a positive treatment outcome more uncertain. This suggests that a patient who is still active in the labor market could be offered surgical treatment regardless of prior non-surgical treatment. An orthopedic surgeon explains:

"*If you take a hard look at it, with a patient whose job is at risk and who needs to return quickly to the labor market, then it nevertheless plays a role for him, whether you choose a treatment with a success rate of 80–90%, which is a "high risk, high reward" treatment (surgery), or you choose to postpone surgery for three months with a treatment (exercise) that might or might not be effective with a lower success rate, with no complications. If you are retired, you are more likely to say: "I would like to try (exercise) because I'm not worried". But if you know you will be fired then it is another matter and you would like the operation here and now.*" (Orthopedic surgeon 1, single interview).

Another example was that of patients who are not motivated for treatment with a more unsure outcome and possibly extended overall treatment time. One orthopedic surgeon said:

"*Most of the patients not motivated for exercise are those still active in the labor market who need to return to their job and thus need a quicker solution.*" (Orthopedic surgeon 1, single interview).

Regardless of any treatment, surgical or non-surgical, a fundamental criterion for treatment success is the patient's motivation for the treatment. Generally, motivation seems to transcend all characteristics that might affect the treatment of choice, making it a relevant point in all patients. One orthopedic surgeon explains:

"*It is a question of whether the patients actually do it (exercise). How compliant they are. Because, you can easily send the patients home and say they should exercise. The question is, however, whether they actually do it. That is the problem*" (Orthopedic surgeon 3, single interview).

Referring a patient to exercise does not mean that the patient complies with this, potentially making the referral a waste of time. This focus on patient adherence to prescribed exercise harmonizes well with the physical therapists' emphasis on adherence. An orthopedic surgeon explains:

"*But there are some patients who say, "I don't want to exercise". That is, "I have heard about it and I don't want to exercise, I would like an operation". You can take the horse to water, but you can't make it drink [provided as an analogy]. It might be that I refer the patient to exercise and that I insist on it. But if the patient comes back three months later and says, "I have not been exercising, now I would like an operation", then the recommendations have been met, but you have wasted three months of both his and my time.*" (Orthopedic surgeon 1, single interview).

Also, a patient who is not adhering to exercise prior to surgery, could have difficulties completing the necessary rehabilitation following surgery. This could make the orthopedic surgeon reconsider the decision of whether surgery is the right treatment for the patient. At the same time, exercise therapy is a "safe" treatment (low risk of complications) that could reduce symptoms, and at the same time the patient is not rushed into a decision about surgery. One orthopedic surgeon explains:

"*So, for me it (exercise) is also a means to see if we should use surgery. But sometimes it is just as much a means to evaluate their motivation. Because if they don't want to exercise at all then their expectations following the operation must be re-assessed. So, I also use it (exercise) as an analysis of their personality. Also, in relation to exercise following surgery.*" (Orthopedic surgeon 4, single interview).

Thus, motivation is an important patient characteristic for the orthopedic surgeons, as they do not want to refer patients to a treatment that they are not going to adhere to due to lack of motivation. This would be a waste of everyone's time and resources and becomes a potential barrier for referring patients to exercise.

**Sub-theme 7: Different purposes of referring a patient to exercise.** The single interviews showed that there is large variation in how exercise is used as a treatment modality among the orthopedic surgeons. Some of the orthopedic surgeons use exercise therapy as a treatment and assess the result on knee-related symptoms, while others use exercise therapy as a means of assessing patient resources and motivation for exercise or to provide them with a "breathing space", where they can consider the possibility of surgery. This was evident form statements like this:

"*. . .when you get the impression that the patient is not completely aware of what it (surgery) involves, then exercise provides a breathing space and it might also help. . .*" (Orthopedic surgeon 1, single interview).

Another view on exercise is that it is a treatment where the patient "can be parked" until surgery is needed. One orthopedic surgeon explained:

"*It (exercise) can be an advantage. If I don't think they are candidates for surgery, then I "park" them out there (to exercise).*" (Orthopedic surgeon 3, single interview).

A general point made by all the orthopedic surgeons is that exercise comes with a low risk of complications and is therefore worth trying when there is uncertainty concerning whether to proceed with surgery. The citation below illustrates this:

"*It is worth a try to postpone surgery three months and try exercise. It might help, and it might not. . .It also worth a try if you can avoid surgery and the associated complications.*" (Orthopedic surgeon 1, single interview).

Assessing the effect of exercise on knee-related symptoms, evaluating patient resources, and providing patients with a "breathing space" to consider the treatment option of surgery all become potential facilitators for referring a patient to exercise.

Essentially, the orthopedic surgeons act as gate-keepers for the different treatment options. The decision of whether patients eligible for KR are referred to exercise prior to potential surgery relies heavily on the preferences and clinical judgement of the orthopedic surgeon. An orthopedic surgeon stated:

"*That is probably the biggest influence we have. To select the patients.*" (Orthopedic surgeon 1, single interview).

In summary, results from our single interviews with the orthopedic surgeons show that adhering to clinical guideline recommendations—and at the same time using clinical expertise and considering patient preferences—creates a professional dilemma among the orthopedic surgeons. On the one hand, facilitators such as *using exercise as a means to examine patient's motivation for rehabilitation*, *providing patients with a low-risk-of-complications treatment while considering the option of surgery* and *knowledge of the effect of exercise can help guide the decision of surgery* support the use of exercise as a treatment modality among orthopedic surgeons for patients with severe knee OA. On the other hand, barriers among the orthopedic surgeons towards referring patients with severe knee OA to exercise were *skepticism towards the effect of exercise and especially the long-term effect in patients with severe knee OA* and *the dilemma of referring patients to exercise who are not motivated for this treatment modality*. These barriers challenge the orthopedic surgeons creating ambivalence in their professional role.

### Theme 3: Coordinated non-surgical and surgical care

Orthopedic surgeons and physical therapists are preoccupied with different aspects of coordinated non-surgical and surgical care. The orthopedic surgeons focus on what kind of treatment they refer to, while the physical therapists focus more on the care pathway as a whole.

**Sub-theme 1: Orthopedic surgeons' skepticism to the content of the exercise treatment they refer to.** A general skepticism concerning referring patients to exercise is that the orthopedic surgeons experience a large variation in the type of exercise intervention the patients are offered following referral. An orthopedic surgeon explains:

"*But you hear a lot of different things. Some have had very good exercise treatment while others have had massage for three months, which might be nice, but I doubt that it helps with their osteoarthritis symptoms.*" (Orthopedic surgeon 1, single interview).

Another barrier to referring patients to exercise is the time required to inform patients about exercise. An orthopedic surgeon says:

"*But there's a lot of information in this (referral to exercise). It is a lot easier to schedule the patients for surgery. It requires a lot more to information to send them (the patients) to training.*" (Orthopedic surgeon 4, single interview).

An advantage of referring patients to the QUADX-1 exercise intervention is that the orthopedic surgeons know exactly what the intervention comprises. An orthopedic surgeon explains:

"*The advantage of this (the QUADX-1 intervention) is that you know what you get. We cannot refer directly to other exercise treatments where we know the content.*" (Orthopedic surgeon 1, single interview).

Referring to exercise without knowing the content of the treatment provided and the time associated with referral to exercise becomes potential barriers for the orthopedic surgeons to refer patients to exercise.

**Sub-theme 2: Responsibilities in coordinated care and engagement in the care pathway.** When an orthopedic surgeon refers a patient to non-surgical treatment then "the surgeon's work is done". All practicalities related to coordinating treatment across sectors should be managed by secretaries, according to the orthopedic surgeons. An orthopedic surgeon says:

"*Well, I just assess the patient and refer them to exercise and they get a new appointment three months later. Then I don't do anything more.*" (Orthopedic surgeon 3, single interview).

And

"*The secretary has to keep track of it (the referral to exercise in the municipality). I should do as little as possible in that respect.*" (Orthopedic surgeon 3, single interview).

Once an orthopedic surgeon has referred a patient to exercise then the physical therapist has responsibility for the treatment. An orthopedic surgeon explains:

"*In principle it is the responsibility of the physical therapist that the exercise intervention is completed.*" (Orthopedic surgeon 3, single interview).

The physical therapists are positive towards coordinated non-surgical and surgical treatment as they believe the patients are provided with an altogether better care pathway when exercise is tried before the decision for surgery is made. This was shown in statements like the following:

"*I like the idea—that the patient isn't told at the first consultation that "you need a knee replacement"–that exercise is tried and then the need for surgery is re-evaluated. That is, "this (exercise) worked for me", or "this did not work for me". I think this is a reasonable care pathway.*" (Physical therapist 4, focus group interview).

The physical therapists also believe that patients would appreciate such a coordinated care pathway and would feel confident that all treatment options have been explored. Further, in

such a coordinated care setting the patients would experience that the health care professionals at the hospital and in the municipality are communicating and working together. A physical therapist explains:

"...I think that the individual patient will feel that they have been taken good care of... That all treatment options have been tried out and that they have had a good care pathway... Also, in relation to communication, that they (the patients) experience that we and they (the orthopedic surgeons at the hospital) are 'on the beat'. That it is a transparent care pathway." (Physical therapist 6, focus group interview).

The physical therapists see another advantage of the coordinated care. Patients that postpone surgery and continue with exercise can be helped in the municipality to maintain their exercise program and can be guided to local gyms and other activities. A physical therapist explains:

"If all care and treatment took place at the hospital, then it could be even more difficult (for the patients) to continue (to exercise) in a local gym... The municipalities are less institutionalized compared to hospitals, and we have better knowledge of the local exercise options. I think this is an advantage." (Physical therapist 2, focus group interview).

Clear allocation of responsibility in relation to exercise referral and providing patients an optimized and transparent care pathway becomes potential facilitators for the coordinated non-surgical and surgical care pathway.

In summary, the orthopedic surgeons express frustration with variation in the treatment provided for the patients when they refer them to exercise in the municipality which becomes a barrier to referring patients to exercise. The physical therapists are positive in respect of the coordinated care pathway as they believe this will mean that patients are provided with quality care. This becomes a facilitator for coordinated non-surgical and surgical treatment.

## Discussion

This study applied a thematic analysis to identify facilitators and barriers among orthopedic surgeons and physical therapists towards a coordinated care pathway of non-surgical and surgical treatment for patients eligible for KR using pre-operative home-based exercise with *one* exercise. Three interrelated themes were found important for whether coordinated surgical and non-surgical care of home-based exercise therapy with *one* exercise was perceived as a facilitator or barrier: 1) *Physical therapists' dilemma with* one *home-based exercise*, 2) *Orthopedic surgeons' dilemma with exercise*, and 3) *Coordinated non-surgical and surgical care.* How coordinated non-surgical and surgical care with home-based exercise therapy with *one* exercise challenges both professions and create ambivalence in their professional roles will be discussed first. Then the orthopedic surgeons' view on exercise and finally the two professions different focus in coordinated non-surgical and surgical care will be discussed.

For the physical therapists, exercise therapy is a central treatment modality characterizing their profession [48,49]. Traditionally, exercise therapy is provided at supervised group sessions allowing physical therapists to monitor treatment closely and adjust accordingly. Reimbursement or self-payment of lifelong supervised exercise is not realistic for most patients making self-management critical. The Osteoarthritis Research Society International (OARSI) recommend patient self-management in their guidelines for non-surgical treatment of patients with knee OA [7]. Providing patients with tools for self-management enables patients to adjust home-based exercise and activities of daily living accordingly (knee pain and symptoms)

thereby increasing the chance of maintained successful treatment [11,12]. The physical therapists interviewed in the present study recognize the importance of providing patients with tools to self-manage their condition and acknowledge the benefits of home-based exercise therapy. Interestingly, this aspect of educating patients in self-management is to some degree in conflict with the physical therapists' engaged treatment approach and strong physical therapist-patient relationship. Through self-management education, physical therapists educate patients to manage their condition without supervision from a physical therapist, making themselves expendable. Their acceptance could be explained by their professional role and the concomitant ethical responsibility to do what is best for the patient [49]. This acknowledgement of supporting self-management in patients becomes a facilitator for potential clinical implementation of the simplified treatment approach.

Both professions are educated to become autonomous professional practitioners [23,48,49]. The sociologist Wilensky defines (healthcare) professionals as: *professionals make inferences; they treat individual clients, make specific decisions, analyze specific cases, or give specific advice on the basis of learned, abstract insights* [55], meaning that a professional has acquired standardized skills enabling them to apply knowledge and treat cases (e.g. patients) [56]. Without this premise, anyone could perform professional work [55]. With this perspective, it is understandable why the physical therapists experience home-based exercise therapy with *one* exercise as a challenge to their professional role. Their normal practice consists of going from a lot of supervision and guidance towards less, as the patients become more independent. Even though the physical therapists distance themselves from the patients, they keep some control over the patient's treatment [33,57]. In the single exercise model, this practice is replaced with only one supervised session after which the patient alone is responsible for the exercise therapy. Home-based exercise therapy with *one* exercise limits the physical therapists' possibility to advise and adjust treatment along the way, which means that they do not have the same control over the treatment as usual. The experience of loss of control challenges their profession role and becomes a barrier for implementing home-based exercise therapy with *one* exercise.

"The modern professional role" as described by Abbott [58] further supports the physical therapists' skepticism towards a less engaged and simplified professional role and treatment approach. Abbott views professions as knowledge systems and describes professional work in three parts: 1) classification of problems combined with information on the specific case and professional expertise, 2) translation of professional expertise to potential solutions for the individual case and 3) to reason and specify the solution into one treatment. In the profession of physical therapy, the first part is comparable to information from a patient's medical history, which is supplemented with a clinical examination, while the second and third part overlap in the process of clinical reasoning and treatment [48,49]. In this perspective, a simplified treatment approach could limit physical therapists´ use of their professional knowledge and expertise. This challenges the physical therapists, as it confronts their professional skills, clinical reflection and reasoning, which become less important as the treatment has been defined beforehand by others [49]. This is in line with previously reported barriers among physical therapists towards home-based exercise therapy with *one* exercise, i.e. "too simplistic" and "restricted intervention, wanted to add manual therapy as a treatment option" and "the role of on-going support" [32].

When an orthopedic surgeon considers surgery for knee OA for a patient who has *not* tried exercise therapy, has strong knee OA symptoms and meets the clinical indication for surgery (i.e. state of the joint, pain and functional disability), then the orthopedic surgeons' professional role is challenged. According to clinical guidelines [7,8], the patient should be referred to exercise therapy to see if this treatment is effective in reducing symptoms. This could conflict with the orthopedic surgeon's opinion of what would be the right treatment, for example

due to a skepticism towards the long-term effectiveness of exercise. The orthopedic surgeon could argue that surgery should be offered right away since the indication for surgery is met regardless of whether the patient has tried non-surgical treatment or not. In this situation, the preferences of the orthopedic surgeon become a barrier for adherence to clinical guidelines (including referral to exercise therapy).

Other challenges for the orthopedic surgeons are their role as agents for the healthcare system. An agent is understood as "one who chooses as the patients themselves would choose if they possessed the information that the orthopedic surgeon does" [59]. Orthopedic surgeons are specialists in orthopedic surgery, and on the one hand they must adhere to clinical guideline recommendations of, for example, exercise therapy. On the other hand, they have to offer the treatment they professionally believe is the correct treatment for that patient at that time (fulfills patient preferences), for example, surgery. Orthopedic surgeons are members of the medical profession, operating under the 'medical pledge', and are thus primarily taking care of the patient's preferences rather than what clinical guidelines prescribe [60]. Thereby, they appear not only to be agents but to be double agents. That is, they are obliged to follow clinical guideline recommendations and at the same time take the patient's preferences into account while also using their clinical expertise. Further, they also need to ensure that a certain number of patients are referred to surgery as the department is economically dependent on the number of surgeries completed. This double agent position can become a barrier for referring patients to exercise therapy. These barriers may not only be relevant for the present study. They may provide new targets for future implementation studies of current guidelines concerning the non-surgical treatment of patients considered for surgery in general.

Another central finding from the study is how orthopedic surgeons expressed skepticism towards (long-term) effect of exercise as a treatment modality in patients with severe knee OA. This skepticism becomes a barrier for referring patients to exercise therapy and a facilitator for referring patients directly to surgery. This finding is supported by data from a study where, if an orthopedic surgeon had a "lack of belief in physical therapy when there is an obvious loss of cartilage" this was associated with fewer referrals to physical therapy [27]. Some orthopedic surgeons question the long-term effect of exercise therapy, which may relate to how they are used to thinking about intervention and effect, that is, they are used to long-lasting effects of their single intervention (surgery). A one-time intervention with exercise therapy (one single session) will have no long-lasting effect without being supplemented with more sessions [61,62]. This is no different from administering a single dose of insulin to a diabetic, where a long-lasting effect is not expected unless supplemented with further doses of insulin [63]. What also makes the insulin-analogy relevant is that the intervention requires behavioral changes in patients, which is also the case for patients with knee OA, who do not necessarily exercise regularly [64]. Taken together, different professions may have different opinions and understandings of the short- and long-term effects of the different interventions that they use to treat patients.

Despite skepticism towards the effect of exercise, the orthopedic surgeons recognize the value of exercise as a treatment modality in their clinical work. For patients in doubt of whether to undergo surgery, a less invasive treatment option is welcomed while the patients consider the option of surgery and associated risks. This recognition of exercise as an option to provide patients with a non-invasive treatment while considering surgery and as a means to supplement the indication for surgery becomes a facilitator for referring patients to exercise.

The recognition of exercise among the orthopedic surgeons also relates to different considerations when referring patients to exercise. Social support and health beliefs besides disease severity are important in the decision of which treatment to proceed with [23]. Gossec et al. found that severity of pain and functional disability could not distinguish between those who

were or were not recommended for KR in patients with radiographically verified hip or knee OA [25]. Similarly, Skou et al. found similar pain levels in patients deemed eligible and not eligible for KR [13]. The decision concerning referral to exercise relies heavily on the preferences of the orthopedic surgeon, that is, whether exercise is used to assess the treatment effect on knee OA-related symptoms, assessing patient resources and motivation, or as a "breathing space" where the patient can consider the possibility of surgery. Referral practice among orthopedic surgeons has previously been reported to vary depending on the level of professional experience [65]. Essentially, the orthopedic surgeons act as gate-keepers for the different treatment options, and this role as gate-keeper becomes both a facilitator and a barrier depending on the purpose for referring or not referring patients to exercise.

In respect of the coordinated care, the main barrier expressed by the orthopedic surgeons was the variation in treatment received by patients in the municipality, as previously reported [27]. An advantage of the QUADX-1 intervention for the orthopedic surgeons was that they knew what the content of the exercise intervention was. In relation to the coordinated care pathway the orthopedic surgeons mainly focus on their own role, e.g. scheduling patients for surgery or referring to non-surgical care. Conversely, the physical therapists focus more on the care pathway as a whole. It becomes a facilitator for the physical therapists to contribute to the coordinated care pathway by providing exercise before the decision on surgery is taken. Optimized coordination of non-surgical and surgical care provides patients with a more comprehensive care pathway exhausting non-surgical treatment options before potential surgery. This may change the care pathway for some patients with postponed surgery, while shortening surgical waiting time for other patients in greater need of surgical care.

In summary, the model of coordinated non-surgical (home-based exercise therapy with *one* exercise) and surgical treatment challenges both the physical therapists and the orthopedic surgeons and creates an ambivalence in their professional roles in different ways. This ambivalence causes mixed feelings and sometimes conflicting interest in the two professions about how to handle home-based exercise therapy with *one* exercise as evidenced by the different facilitators and barriers in the two professions. Thus, even though home-based exercise therapy with *one* exercise is described and discussed as a standardized and practically simple intervention [36] the results from the study show that the intervention is perceived as complex in the context of clinical interprofessional coordination of treatment [66,67]. The intervention is assigned a situated meaning from the two professions and is not a uniform object shown by the different facilitators and barriers. This is important knowledge for future implementation of the intervention, where it is not uncommon to think of the implementation as standardized leading to a "one size fits all" implementation approach. When implementing the intervention, it must be considered "complex", where several implementation strategies must be used (a multifaceted approach) [68]. Further, the intervention must be adapted to both professions, that is, the situated meaning and the facilitators and barriers the physical therapists and orthopedic surgeons experience.

## Strengths and limitations

This study has both strengths and limitations which should be considered when interpreting the findings. One of the main strengths is that all potential participants were included for the interviews. That is, all physical therapists and eligible orthopedic surgeons (supervisors and co-authors deemed not eligible due to conflict of interest) involved in the QUADX-1 trial participated in the interviews. Further, the study involves both groups of healthcare professionals that coordinate treatment for patients with severe knee OA in their daily clinical practice (i.e. physical therapists and orthopedic surgeons). The study is reported according to the Standards

for Reporting Qualitative Research: A Synthesis of Recommendations (SRQR) checklist [69], which we believe strengthens the transparency and validity of the study.

The main limitations are that despite including all potential participants for the interviews, we only interviewed six physical therapists and four orthopedic surgeons. It is unknown if additional participants would have added new perspectives or further supported the findings of the study. The use of two different interview approaches (focus group- and single interviews) could have introduced bias. By only applying single interviews for the orthopedic surgeons, we missed the interactions created during focus group interviews. Conversely, in single interviews the participant is not at risk of being overwhelmed or ignored by more dominant participants. Though interview data from the orthopedic surgeons was collected on an individual basis the same themes were present among the interviewed orthopedic surgeons [47]. The primary investigator's (RSH) professional background is physical therapy. His preconception of both the physical therapy and orthopedic surgery professions could have affected his conducting of interviews, data analysis and interpretation of the results. To accommodate this, all steps were discussed with and approved by co-authors (TB and JK) during the process and the final results were approved by all co-authors (i.e. physical therapists, an orthopedic surgeon and a nurse).

## Conclusion

We found that both physical therapists and orthopedic surgeons were challenged by coordinated non-surgical and surgical treatment of patients eligible for KR using pre-operative home-based exercise therapy with *one* exercise as evidenced by the identified facilitators and barriers. The intervention created ambivalence in the professional role of both the physical therapists and orthopedic surgeons. The physical therapists were skeptical about over-simplified exercise therapy but positive towards patient self-management. The orthopedic surgeons were skeptical about the potential lack of a long-term effect of exercise therapy in patients with severe knee OA but acknowledged exercise therapy as an alternative treatment option in daily clinical practice. This ambivalence in the professional role is important to consider when planning implementation of the intervention as it may appear simple but is regarded as complex.

## Supporting information

**S1 File. Standards for Reporting Qualitative Research (SRQR) checklist.**
(DOCX)

**S2 File. Interview guide physical therapists.**
(PDF)

**S3 File. Interview guide orthopedic surgeons.**
(PDF)

**S4 File. Adjustments and additions to the semi-structured interview guides.**
(PDF)

**S5 File. Audit trail of the thematic analysis and a section on trustworthiness.**
(PDF)

**S6 File. Written informed consent form.**
(PDF)

**S7 File. Research data 1—Anonymous transcribed focus group interview with physiotherapists (in original language; Danish).**
(PDF)

**S8 File. Research data 2—Anonymous transcribed single interviews with orthopedic surgeons (in original language; Danish).**
(PDF)

## Acknowledgments

We gratefully thank all participating physical therapists and orthopedic surgeons for their contribution to the present study.

## Author Contributions

**Conceptualization:** Rasmus Skov Husted, Thomas Bandholm, Michael Skovdal Rathleff, Anders Troelsen, Jeanette Kirk.

**Data curation:** Rasmus Skov Husted, Jeanette Kirk.

**Formal analysis:** Rasmus Skov Husted, Thomas Bandholm, Michael Skovdal Rathleff, Jeanette Kirk.

**Funding acquisition:** Thomas Bandholm, Anders Troelsen.

**Investigation:** Rasmus Skov Husted, Thomas Bandholm, Jeanette Kirk.

**Methodology:** Rasmus Skov Husted, Jeanette Kirk.

**Project administration:** Rasmus Skov Husted, Thomas Bandholm.

**Supervision:** Thomas Bandholm, Michael Skovdal Rathleff, Anders Troelsen, Jeanette Kirk.

**Writing – original draft:** Rasmus Skov Husted.

**Writing – review & editing:** Rasmus Skov Husted, Thomas Bandholm, Michael Skovdal Rathleff, Anders Troelsen, Jeanette Kirk.

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
