## [Decision Letter · Decision Letter 0]

17 Apr 2020

PONE-D-20-02454

Perceived facilitators and barriers among physical therapists and orthopedic surgeons to pre-operative home-based exercise with one exercise-only in patients with severe knee osteoarthritis: A qualitative interview study nested in the QUADX-1 trial

PLOS ONE

Dear Dr Husted,

Thank you for submitting your manuscript to PLOS ONE. After careful consideration, we feel that it has merit but does not fully meet PLOS ONE’s publication criteria as it currently stands. Therefore, we invite you to submit a revised version of the manuscript that addresses the points raised during the review process.

Please consider the reviewers comments and my comments below. 

We would appreciate receiving your revised manuscript by 17 May 2020. To enhance the reproducibility of your results, we recommend that if applicable you deposit your laboratory protocols in protocols.io, where a protocol can be assigned its own identifier (DOI) such that it can be cited independently in the future. For instructions see: http://journals.plos.org/plosone/s/submission-guidelines#loc-laboratory-protocols

We look forward to receiving your revised manuscript.

Kind regards,

Andrew Soundy

Academic Editor

PLOS ONE

Additional Editor Comments (if provided):

Design: Please give a reference for the design and identify the ontology and epistemology of this design. This will impact comments for the results in the next version.

Make sure you adhere to the methods section of a framework like COREQ (Tong) or SRSQ (Obrien) – from these consider the following:

Sample size.

Use a supplementary file to show changes in interview schedule – was this also a cognitive interview?

Identify what literature was used to develop the interview schedule

An audit trail must be given for your thematic analysis (examples of each stages of the analysis you say has been done so it can be checked – have this in a supplementary file – move table 1 to show the development of this.)

Have a section at the end on trustworthiness and how this was identified. Make sure this links to your paradigmatic position and doesn’t contradict it.

Could the analysis have been driven by apriori analysis? Explain your answer

2. In your Methods section, please provide additional information about the participant recruitment method and the demographic details of your participants. Please ensure you have provided sufficient details to replicate the analyses such as: a) the recruitment date range (month and year), b) a description of any inclusion/exclusion criteria that were applied to participant recruitment, c) a table of relevant demographic details, and d) descriptions of where participants were recruited (the specific locations and names of the municipalities and hospitals) and where the research took place.

3. Thank you for clarifiying that oral consent was obtained from the participants for the interviews condcuted in this study. In your Methods section, please also state:

- Why written consent could not be obtained

- Whether consent was informed

- Whether the Institutional Review Board (IRB) approved use of oral consent

- How oral consent was documented

For more information, please see our guidelines for human subjects research: https://journals.plos.org/plosone/s/submission-guidelines#loc-human-subjects-research

4. We note you have included a table to which you do not refer in the text of your manuscript. Please ensure that you refer to Table 2 in your text; if accepted, production will need this reference to link the reader to the Table.

"The authors have declared that no competing interests exist.

Dr. Troelsen reports personal fees from Zimmer Biomet, grants from Zimmer Biomet, personal fees from European Knee Society, outside the submitted work."

Reviewers' comments:

Reviewer's Responses to Questions

**Comments to the Author**

1. Is the manuscript technically sound, and do the data support the conclusions?

Reviewer #1: No

2. Has the statistical analysis been performed appropriately and rigorously? 

Reviewer #1: N/A

3. Have the authors made all data underlying the findings in their manuscript fully available?

Reviewer #1: No

4. Is the manuscript presented in an intelligible fashion and written in standard English?

Reviewer #1: Yes

5. Review Comments to the Author

Reviewer #1: Dear authors,

I have read through your article with great interest. I acknowledge the value of exercise in the management of knee osteoarthritis, as well as the need for in-depth understanding of the role healthcare professionals and contextual factors in implementing innovations in clinical practice. After reading your manuscript though, its aims and significance were still unclear to me. I hope my comments below will help showcase the study.

Title

Not in line with aims as stated in abstract. Also, make sure the title reflects the manuscript content.

Introduction

Overall, a better case needs to be made as to why this study is needed. At present the study aims do not logically follow from the arguments made in the introduction. There are important gaps. Specifically:

(1) Why is exercise treatment appropriate for patients eligible for knee replacement?

This is a specific subgroup of people living with knee OA (kOA), typically those who have severe radiographic and/or clinical characteristics. Two guidelines are cited as main supporting literature. The one in English (McAlindon et al 2014), which I could access, firstly is based on systematic reviews of studies from patients with kOA of any severity/ progression; secondly, small short term benefits are reported for land based exercise (including aerobic and range of movement exercises), and similar effect sizes are reported for strengthening exercises. It cannot be concluded that exercise therapy should be offered to pre-surgical kOA patient. More population-specific and up-to-date literature is needed.

The authors could also expand on who is/ is not eligible for KR.

(2) What is the context of clinical practice and healthcare policies in Denmark? What patients’/ healthcare professionals’ needs does the proposed trial meet/ what innovation(s) does it introduce?

These points need to be linked to the components of the proposed intervention components (e.g., advantages of home based exercise, shortcomings of exercise therapy that includes variety of exercises).

Regarding physical therapy, aren’t patients offered this treatment option at an earlier point in disease progression, prior to being eligible for KR? Or the suggested pre-surgical one-exercise treatment is additional to that. It would also be helpful to further explain what constitutes “coordination”. It is described as “novel” (line 144), but it is not clear how it differs from simple referral to exercise therapy by the orthopaedic surgeon (which is the case in UK).

(3) What are the potential advantages/ hypothesized benefits of using one-only exercise therapy specifically?

(4) Other:

Para 2, makes the argument of strong physical therapist-patient relationship in contrast to surgeon-patient relationship before moving on (par 3) to the need for coordination between the two. It is unclear what the purpose of such an argument is, since the intervention under scope is characterized by limited physical therapist- patient interaction.

(5) The aim stated in line 106 (“identify perceived facilitators and barriers”) is different from aim stated in abstract (“investigate key stakeholder perspectives”). In the methods section this qualitative study is presented as aiming at “clinical applicability and implementation” (line 126). Please be clear and consistent about the study aims, which should also be reflected in the interview schedules, results and discussion.

Methods

As the aims of the study remain to be clarified, further revisions might be needed in the process of analysis. For example, if the study focuses on identifying barriers and facilitators to implementation the unit(s) of analysis might change, or a deductive approach could be adopted at some point in the process.

That aside, the main issue here is the limited study sample, which raises questions about credibility of the findings, especially since data saturation is not discussed/ was not achieved. How was variability in participants’ experiences ensured? Some information on relevant participants’ characteristics (e.g. level of experience in their profession) and on hospital and rehabilitation centres (e.g., how were they selected? Considered typical?) could help in this direction. Further, the authors could consider (A) repeating the interviews with the participants post-intervention or (B) interview a wider sample of healthcare professionals outside the QUADX-1 trial, until saturation is achieved. The latter would be particularly useful in planning implementation of the proposed therapy in standard healthcare practice.

Also, a couple of discrepancies: the chosen method of data analysis is described at places as thematic analysis (e.g. lines 219, 234, 253, 264, 710), elsewhere as content analysis (abstract, citation 35). In the surgeons’ interview schedule (S3, Q4) self-management, including education, are mentioned. These are further reported in results and discussion. However, they are not described in the introduction and methods. Please ensure consistency and transparency in reporting.

Other, minor comments:

• Some details of the Context could fit better in the introduction, especially where literature review is involved (e.g., lines 131-133).

• Line 220- was all text divided into meaning units?

• How many municipalities are served by/linked to this orthopaedic department (line 152)?

• Line 178 “experiences…on the coordinated… treatment”. Consider removing the term “experience”, since this is about preconceptions, not actual experiences with the trial (line 196)

• Lines 229-232. More details on how these preconceptions changed, would be useful.

Results

I acknowledge that data analysis was reviewed and agreed upon by more than one researchers. However, the reported results do not appear to clearly reflect (patterns found in) the data and the subthemes and themes appear to be poorly supported. Specifically:

• Quotes do not represent all participants. There are no quotes from physical therapists 4 and 5. Instead quotes from selected participants appear repeatedly. For example, subthemes relevant to surgeons are supported by more than one quote from the same participant but not more than two participants are referenced (subthemes 3, 5, 6, 8, 9).

• The subtheme title and content (i.e., authors description, original quotes) are often not in line and there doesn’t seem to be a connecting thread within each subtheme. In addition, there is overlap between suggested subthemes. Two examples: under the theme “professional role simplified” a substantial amount of text (lines 344-357) discusses physical therapists’ expressed preference towards more than one treatment options; the supporting quote is on therapists’ views regarding good candidates for the proposed treatment. The context of the next subtheme, “skepticism towards one home-based exercise”, includes again a discussion on physical therapists’ suggestions for more treatment options (lines 367-374), positive views of one exercise (376-94) and reference to “professional self-image” (376-78).

• Parts of the results sections read more like discussion of the findings/ authors’ interpretations or review of the literature. Some examples: lines 277-280; 288-292; 452 (citation). On a similar note, the choice of wording in authors’ descriptions at points reads more like their own perspectives rather than participants’ perspectives. Some examples: 586-587 “exercise is used in a less constructive and inclusive way (by the surgeons)”; “professional dilemma” (612)- unclear what the dilemma is on, since surgeons’ quotes overall give their reasoning and clear-cut perspectives for using or not using a treatment (eg quotes on 549, 569, 579); “preferences” (line 605) rather than “clinical judgment”.

• Original data provided is not in English, therefore not accessible to the majority of the readers.

• “Barriers” and “facilitators” are not reported in a consistent way in the results and the relevant research question is not answered comprehensively.

I would recommend that the authors:

• Review the analytical process aiming to create more homogenous and comprehensive subthemes/ themes. Finding should reflect patters found across participants. This might not be that important if the aim is to identify trial implementation strategies, but this need to be clearer and methods modified accordingly

• Provide a supplementary table where under each subtheme there is one quote from each participant that supports the subtheme

• Provide a translation in English of a single interview and part of the focus group

• Be succinct when reporting the findings. Indicate which text represents views/ perspectives etc. expressed by the participants (e.g. participants described, agreed, emphasized/...). Also provide more indications of frequency/ homogeneity of the statements made (eg all/ most/ few physical therapists expressed…). Be more reflective and refrain from making interpretations in this section.

• Also, please keep a consistent and easy to follow structure.

Examples of issues in the structure that are confusing for the reader: “physical therapists” section (line 276 onwards) and orthopedic surgeons (line 407 onwards) appear as subthemes, with their own introductory paragraph, although these are not previously mentioned- unclear what they represent; Subtheme numbering: in lines 254-260 the subthemes are numbered 1 to 9, Table 2 has no numbering, whereas in the remaining results sections subthemes are re-numbered from 1.

Discussion

I believe this section will be revisited following revisions on the previous sections, therefore it won’t be of use commenting on the content as it is. Overall, a number of good points are discussed. Areas for refinement could be: clear take-home messages (3-4) that bring the discussion back to the points raised in the introduction (study aims, study population, existing gaps, concepts, trial implementation etc.).

6. PLOS authors have the option to publish the peer review history of their article (what does this mean?). If published, this will include your full peer review and any attached files.

Reviewer #1: No

---

## [Author Response · Author response to Decision Letter 0]

21 Aug 2020

Point-by-point revision of

PONE-D-20-02454

Perceived facilitators and barriers among physical therapists and orthopedic surgeons to pre-operative home-based exercise with one exercise-only in patients with severe knee osteoarthritis: A qualitative interview study nested in the QUADX-1 trial

Enclosed, please find a revised version of the above manuscript. We have carefully addressed all comments and questions from the reviewer and editor, which we believe has significantly improved the quality of the manuscript. We have submitted a detailed point-by-point list of our replies to the comments and questions made and associated changes made in the manuscript. This has been submitted together with the submission of the revised manuscript. All parts in the text that have been modified are described in the below point-by-point response.

Journal requirements

Journal requirement #1

Please ensure that your manuscript meets PLOS ONE's style requirements, including those for file naming. The PLOS ONE style templates can be found at https://journals.plos.org/plosone/s/file?id=wjVg/PLOSOne_formatting_sample_main_body.pdf and https://journals.plos.org/plosone/s/file?id=ba62/PLOSOne_formatting_sample_title_authors_affiliations.pdf

Author response

Thanks for notifying us on these omissions. We have now updated the style requirements accordingly.

Journal requirement #2

In your Methods section, please provide additional information about the participant recruitment method and the demographic details of your participants. Please ensure you have provided sufficient details to replicate the analyses such as: a) the recruitment date range (month and year), b) a description of any inclusion/exclusion criteria that were applied to participant recruitment, c) a table of relevant demographic details, and d) descriptions of where participants were recruited (the specific locations and names of the municipalities and hospitals) and where the research took place.

Author response

We thank the reviewer for this clarifying comment. We agree with the comment and have added the below information to the manuscript. We refrain from naming the individual municipalities and the hospital to keep the identity of the participants anonymous. We believe that by doing this we also adherence to the General Guidelines for Human Research Participant Data as stated on the PLOS ONE website (https://journals.plos.org/plosone/s/data-availability). The below is an excerpt from the website: Data that are not directly identifying may also be inappropriate to share, as in combination they can become identifying. For example, data collected from a small group of participants, vulnerable populations, or private groups should not be shared if they involve indirect identifiers (such as sex, ethnicity, location, etc.) that may risk the identification of study participants. 

Action taken

The highlighted text has been added to the “recruitment and study participants” and “procedures” paragraphs. Demographic details are provided in Table 1.

We recruited six physical therapists from three municipalities in the capital region of Denmark and four orthopedic surgeons from one university hospital in the capital region of Denmark involved in the QUADX-1 trial [1] (Table 1). Inclusion criteria: participants had to be involved in the QUADX-1 trial, as the intervention under study was not implemented in routine clinical practice. No exclusion criteria were applied. Thus, the six physical therapists and four orthopedic surgeons represents all possible participants as we sought situational representativeness rather than demographic [2]. We recognize that a sample size of ten participants could be a limitation and inadequate to attain data saturation. However, we consider this necessary as this was the only possibility to investigate facilitators and barriers among the orthopedic surgeons and physical therapists involved in the QUADX-1 trial. When interpreting the results this should be kept in mind. Participants were contacted by the primary investigator and interview moderator (RSH) by e-mail with an invitation to participate in the interviews between November 2016 and June 2017. RSH sent the invitations because he would be conducting the interviews. All invited participants accepted. All eligible physical therapists had daily clinical work with patients diagnosed with knee OA and rehabilitation following KR. All eligible orthopedic surgeons had daily clinical work with patients potentially eligible for KR due to knee OA symptoms. A random sample of patients participating in the QUADX-1 trial were also interviewed about their perceptions of facilitators and barriers towards coordinated non-surgical and surgical treatment using pre-operative home-based exercise therapy with one exercise. This work is as yet unpublished.

Changes made at page 8-9, lines 168-188

And

The interviews took place in meeting rooms at a university hospital in the capital region of Denmark.

Changes made at page 11, lines 221-222.

Journal requirement #3

Thank you for clarifying that oral consent was obtained from the participants for the interviews conducted in this study. In your Methods section, please also state:

- Why written consent could not be obtained

- Whether consent was informed

- Whether the Institutional Review Board (IRB) approved use of oral consent 

- How oral consent was documented

For more information, please see our guidelines for human subjects research: https://journals.plos.org/plosone/s/submission-guidelines#loc-human-subjects-research

Author response

We thank the reviewer for this comment. We agree that this is important to make clearer in the manuscript.

Consent was informed. We have added information to the manuscript to make this clearer. The IRB approved the study with no specification on the type of informed consent for qualitative interviews. We choose oral informed consent, why written informed consent was not originally obtained. Written informed consent was not originally obtained for the following reasons: 1) the head of departments approved the health care professional’s participation in the study, 2) we did not expect the interviews to include personally identifiable topics (which they did not) and 3) as the head of departments approved the health care professionals participation and that the participants gave oral consent, informed consent was essentially given twice. However, to make any uncertainties and misunderstandings clear we have gathered written informed consent from the interview participants subsequently. Please see Supplementary Information 6 for an example of the written informed consent formula.

Action taken

The highlighted text has been added to the “Ethics” paragraph.

The study was performed according to the Helsinki Declaration [3]. Before undertaking the interviews, all participants were provided oral and written information about the aim of the study, procedures to be undertaken, potential risks and benefits of participation, expected duration of the study and extent of confidentiality of personal identification, assured that participation was voluntary and that they could withdraw consent at any time during the interview. All invited participants were allowed a minimum of 24 hours to consider participation. All participants were informed about anonymity and confidentiality and gave written informed consent to participate in the interviews. All participants are pseudo-anonymized and reported data are de-identified (no mentioning of names or date of birth). Data were stored on a file drive secured by log-in. The study has been approved by The National Committee on Health Research Ethics (Protocol no.: H-16025136).

Changes made at page 13-14, lines 269-279.

Journal requirement #4

We note you have included a table to which you do not refer in the text of your manuscript. Please ensure that you refer to Table 2 in your text; if accepted, production will need this reference to link the reader to the Table.

Author response

Thanks for spotting this error, which we have now corrected. Table 2 is now table 3, as a new table (Table 1: Participant demographics) has been introduced earlier in the manuscript.

Action taken

A reference for table 3 is now provided in the first paragraph of the results section. 

…and 9) Responsibilities in coordinated care and engagement in the care pathway (Table 3). The themes and sub-themes represent different facilitators and barriers among orthopedic surgeons and physical therapists towards home-based pre-operative exercise in patients eligible for KR.

Changes made at page 14, line 292.

Journal requirement #5

Thank you for stating the following in the Competing Interests section:

"The authors have declared that no competing interests exist.

Dr. Troelsen reports personal fees from Zimmer Biomet, grants from Zimmer Biomet, personal fees from European Knee Society, outside the submitted work."

Author response

Thanks for notifying us on this. We have added the required sentence to the declaration of competing interests. Please see the cover letter.

Additional Editor Comments (if provided)

Editor comment #1

Design: Please give a reference for the design and identify the ontology and epistemology of this design. This will impact comments for the results in the next version.

Author response

We thank the editor for this comment. We believe we have given a reference for the design in the paragraph named Context: The QUADX-1 trial (Page 6, lines 119-136). To make this clearer we have added a sentence and a reference in the Design paragraph (Page 8, lines 152-153). 

Ontologically, clinical practice regarding patients with knee osteoarthritis is understood as dynamic, complex and in constant change and that there is no definitive truth to this context. Knowledge is relational and is created among people. For this reason, it is relevant to investigate and understand facilitators and barriers among physical therapists and orthopedic surgeons towards a model of coordinated non-surgical and surgical care with an exercise intervention designed as home-based with only one exercise. To understand this context better a hybrid I design is applied [4]. A hybrid I design emphasises the effectiveness of an intervention but at the same time recognizes that clinical practice is complex, and that implementation is not a linear process. This is taken into consideration by also investigating the context and the associated facilitators and barriers. 

Epistemologically the study applies single- and focus group interviews as these are acknowledged methods to investigate topics which cannot be quantified or measured. Qualitative methods are interpretive and appropriate to get a deeper understanding of attitudes, behaviour, opinions and feelings and potential discrepancies among these. This is also relevant when it comes to understanding facilitators and barriers among physical therapists and orthopedic surgeons towards a model of coordinated non-surgical and surgical care.

Action taken

We performed interviews to understand the perceived barriers and facilitators among the orthopedic surgeons and physical therapists regarding this novel coordination of surgical and non-surgical treatment prior to the beginning of the parent trial. By using interviews RSH was able to interpret words as well as body language to obtain a better understanding. Further, it provided an opportunity to create trust between RSH and the participants which is important when talking about facilitators and barriers related to work, habits and potential associated feelings. This is an acknowledged design when investigating facilitators and barriers in an implementation context [4].

Changes made at page 7-8, lines 146-153.

Editor comment #2

Make sure you adhere to the methods section of a framework like COREQ (Tong) or SRSQ (Obrien) – from these consider the following: Sample size.

Author response

We thank the editor for this comment. We agree that this is important to highlight in the manuscript. We recognize the limitation of the sample size and the consequences it could have regarding data saturation. The participants were selected for their ability to provide information about the topic under investigation. We sought situational representativeness rather than demographic, andparticipants were selected based on their ability to provide information about the area under investigation [2]. As stated below, we found this limitation necessary to investigate facilitators and barriers among the health care professionals involved in the clinical trial, as these were the only possible participants to the interviews. We have made this potential limitation clearer in the manuscript. We have reported the manuscript using the SRSQ checklist, please see appendix 1.

Action taken

The highlighted text has been added to the “recruitment and study participants” paragraph.

Thus, the six physical therapists and four orthopedic surgeons represents all possible participants as we sought situational representativeness rather than demographic representativeness (37). We recognize that a sample size of ten participants could be a limitation and inadequate to certify data saturation. However, we consider this necessary as this was the only possibility to investigate facilitators and barriers among the orthopedic surgeons and physical therapists involved in the QUADX-1 trial. When interpreting the results this should be kept in mind.

Changes made at page 8-9, lines 172-178.

Editor comment #3

Use a supplementary file to show changes in interview schedule – was this also a cognitive interview?

Author response

We thank the editor for this comment. We agree and believe that the supplementary file adds to the transparency of the methods used. This was not a cognitive interview.

Action taken

We have added a supplementary file with adjustments and additions to the semi-structured interview guides. Please see supplementary file S4. 

Following every interview, RSH and JK listened to the audio file and adjusted the interview guide based on new important topics (S4 File).

Changes made at page 11, lines 234-236.

Editor comment #4

Identify what literature was used to develop the interview schedule.

Author response

We thank the editor for this comment. We agree that highlighting the literature used to develop the interview guides provides transparency to the foundation and design of the study.

The interview guides were based on literature related to the organization and recommended order of non-surgical and surgical treatment in patients eligible for knee replacement [5–9], effectiveness of exercise therapy in patients with knee OA [10–13], traditional organization of exercise therapy [14–16], knee replacement surgery [8,9,17], previous identified barriers and facilitators towards exercise therapy and surgery [18–20] and shared decision-making [21] in patients eligible for knee replacement.

Action taken

The above information has been added to the paragraph “Interviews: Focus group and single interviews” in the manuscript. The paragraph has been changed from:

Both the focus group interview and the single interviews were guided by semi-structured interview guides with open-ended questions (S2 File and S3 File). A “funnel approach” was used at all interviews starting with broad open-ended questions followed by probing and sensitizing questions aiming to elicit deeper and more detailed information [22].

To:

Both the focus group interview and the single interviews were guided by semi-structured interview guides with open-ended questions (S2 File and S3 File). The interview guides were based on literature related to the organization and recommended order of non-surgical and surgical treatment in patients eligible for knee replacement [5–9], effectiveness of exercise therapy in patients with knee OA [10–13], traditional organization of exercise therapy [14–16], knee replacement surgery [8,9,17], previous identified barriers and facilitators towards exercise therapy and surgery [18–20] and shared decision-making [21] in patients eligible for knee replacement. A “funnel approach” was used at all interviews starting with broad open-ended questions followed by probing and sensitizing questions aiming to elicit deeper and more detailed information [22]. 

Changes made at page 10-11, lines 205-215.

Editor comment #5

An audit trail must be given for your thematic analysis (examples of each stages of the analysis you say has been done so it can be checked – have this in a supplementary file – move table 1 to show the development of this).

Have a section at the end on trustworthiness and how this was identified. Make sure this links to your paradigmatic position and doesn’t contradict it.

Author response

We thank the editor for this comment. We agree with the comment and believe that providing an audit trail will strengthen the transparency of the steps taken in the thematic analysis. We have added an audit trail and a section on trustworthiness as supporting information. 

Action taken

The following reference to the supporting information has been added to the manuscript.

An audit trail of the thematic analysis and a section on trustworthiness is provided as supporting information (S5 file).

Changes made at page 12, lines 260-261.

Editor comment #6

Could the analysis have been driven by a priori analysis? Explain your answer.

Author response

Yes, the analysis could have been driven by an a priori analysis. If we had had a hypothesis which we wanted to test or had applied a deductive analysis based on a framework or theory the analysis could have been driven by an a priori assumption [23,24]. As we applied inductive analysis with an exploratory aim the analysis was not driven by an a priori assumption [25]. 

Reviewer comments

Reviewer #1: 

Dear authors, I have read through your article with great interest. I acknowledge the value of exercise in the management of knee osteoarthritis, as well as the need for in-depth understanding of the role healthcare professionals and contextual factors in implementing innovations in clinical practice. After reading your manuscript though, its aims and significance were still unclear to me. I hope my comments below will help showcase the study.

Reviewer comment #1

Title

Not in line with aims as stated in abstract. Also, make sure the title reflects the manuscript content.

Author response

We thank the reviewer for this comment. We agree that the title and aims stated in the abstract and manuscript were not aligned. We have adjusted the title and aligned the aims stated in the abstract and manuscript. We have left the information on “coordinated non-surgical and surgical treatment” out of the title to make it shorter and more reader friendly. 

Action taken

The title has been changed from:

Perceived facilitators and barriers among physical therapists and orthopedic surgeons to pre-operative home-based exercise with one exercise-only in patients with severe knee osteoarthritis: A qualitative interview study nested in the QUADX-1 trial.

To:

Perceived facilitators and barriers among physical therapists and orthopedic surgeons to pre-operative home-based exercise with one exercise-only in patients eligible for knee replacement: A qualitative interview study nested in the QUADX-1 trial

Changes made at page 1, lines 2-4.

The aim in the abstract has been changed from:

This study aimed to investigate key stakeholder perspectives on pre-operative, home-based exercise therapy with one exercise-only in patients eligible for KR.

To:

The aim of this study was to identify perceived facilitators and barriers – among orthopedic surgeons and physical therapists – towards coordinated non-surgical and surgical treatment of patients eligible for knee replacement using pre-operative home-based exercise therapy with one exercise. 

Changes made at page 2, lines 28-31.

The aim in the manuscript has been changed from:

The aim of this study was to identify perceived facilitators and barriers – among orthopedic surgeons and physical therapists – towards coordinated non-surgical and surgical treatment of patients with severe knee osteoarthritis using pre-operative home-based exercise therapy with one exercise. 

To:

The aim of this study was to identify perceived facilitators and barriers – among orthopedic surgeons and physical therapists – towards coordinated non-surgical and surgical treatment of patients eligible for knee replacement using pre-operative home-based exercise therapy with one exercise. 

Changes made at page 6, lines 110-113.

Introduction

Overall, a better case needs to be made as to why this study is needed. At present the study aims do not logically follow from the arguments made in the introduction. There are important gaps. Specifically:

Reviewer comment #2

Why is exercise treatment appropriate for patients eligible for knee replacement?

This is a specific subgroup of people living with knee OA (kOA), typically those who have severe radiographic and/or clinical characteristics. Two guidelines are cited as main supporting literature. The one in English (McAlindon et al 2014), which I could access, firstly is based on systematic reviews of studies from patients with kOA of any severity/ progression; secondly, small short term benefits are reported for land based exercise (including aerobic and range of movement exercises), and similar effect sizes are reported for strengthening exercises. It cannot be concluded that exercise therapy should be offered to pre-surgical kOA patient. More population-specific and up-to-date literature is needed.

The authors could also expand on who is/ is not eligible for KR.

Author response

We thank the reviewer for this clarifying comment. We agree with the comment and have added the following to the manuscript, to make the argument for pre-operative exercise therapy for patients eligible for knee replacement clearer in the manuscript. We have also added a paragraph on who is eligible for knee replacement.

Action taken

The highlighted text has been added to the introduction paragraph. 

Both international and national guidelines recommend non-surgical treatment (i.e. exercise therapy, weight loss, self-management and education) before surgery is considered in patients eligible for KR [5–7,26–28] – and recent studies show that exercise therapy can provide clinically relevant improvements in knee OA symptoms in patients eligible for KR [10–13,29,30]. Despite this, it is estimated that up to 25% of patients could be inappropriately receiving KR prior to the recommended non-surgical treatment [17].

Changes made at page 4, lines 66-71.

Eligibility for KR is based on several factors including; the patient’s medical history (i.a. knee pain, quality of life, limitations in daily living), physical examination (i.a. active and passive range of motion, palpation), effect of previous non-surgical treatment [5,6,26] and x-rays [21,31]. Knee OA is roughly categorized as mild-to-moderate or severe and the main difference between the two categories is the level of symptomatology (e.g. knee pain levels) [32]. Patients with severe knee OA are more likely to be deemed eligible for KR and undergo surgery [33–35]. Most patients referred to KR present with severe radiographic knee OA (i.e. clear joint space narrowing, osteophytes, sclerosis and joint deformity) as this is believed to be a good indication as to whether KR will be an effective treatment [36].

Changes made at page 4, lines 69-77.

Reviewer comment #3

What is the context of clinical practice and healthcare policies in Denmark? What patients’/ healthcare professionals’ needs does the proposed trial meet/ what innovation(s) does it introduce?

These points need to be linked to the components of the proposed intervention components (e.g., advantages of home based exercise, shortcomings of exercise therapy that includes variety of exercises).

Author response

We thank the reviewer for this comment. We agree that the context of the study is important. The main challenges in Danish Healthcare is coordination of non-surgical and surgical care as many patients consult an orthopedic surgeon without having tried non-surgical care. This is not in line with national recommendations and a waste of resources. This trial investigates a potential solution for this challenge. The purpose of the investigated coordinated care pathway of non-surgical and surgical treatment is to improve care provided to patients eligible for knee replacement. The context of clinical practice is that currently there is no coordinated evaluation of the effect of non-surgical treatment. That is, when an orthopedic surgeon refers a patient to non-surgical treatment before surgery (complying with national guidelines and recommendation) no coordinated evaluation exists where a healthcare professional evaluates whether this treatment has been sufficient or whether a new treatment should be commenced, e.g. surgery. We believe we have described this on page 5, lines 78-82 – first with an example of an existing coordinated care pathway and then with an example of a lacking coordinated care pathway. On page 6, lines 121-123 the coordinated treatment is described in short again. The purpose of offering one home-based exercise is to provide patients with a simple and cheap alternative to group-based exercise (often with self-payment).

The components and content of the exercise intervention (one home-based knee-extensor exercise) are independent or the coordination of care. That is, in relation to the coordinated care pathway of non-surgical and surgical treatment the exercise intervention could also have been supervised with several exercises. The purpose of the coordinated care pathway with re-evaluation of the non-surgical treatment by an orthopedic surgeon would be the same. 

Reviewer comment #4

Regarding physical therapy, aren’t patients offered this treatment option at an earlier point in disease progression, prior to being eligible for KR? Or the suggested pre-surgical one-exercise treatment is additional to that. 

Author response

We thank the reviewer for this comment. Yes, according to guidelines and recommendations patients should be referred to exercise therapy at an earlier stage in the disease progression – which most patients are. Despite this, some patients still consult an orthopedic surgeon without have tried out exercise [37]. This is a big challenge in the Danish healthcare system. At this point the pre-operative home-based exercise intervention with one exercise is introduced as a simple and cheap alternative to supervised exercise at fixed time point at a rehabilitation centre (often with self-payment).

Reviewer comment #5

It would also be helpful to further explain what constitutes “coordination”. It is described as “novel” (line 144), but it is not clear how it differs from simple referral to exercise therapy by the orthopaedic surgeon (which is the case in UK).

Author response

The context of clinical practice is that currently there is no coordinated evaluation of the effect of non-surgical treatment. That is, when an orthopedic surgeon refers a patient to non-surgical treatment before surgery (complying with national guidelines and recommendation) no coordinated evaluation exists where a healthcare professional evaluates whether this treatment has been sufficient or whether a new treatment should be commenced, e.g. surgery. We believe we have described this on page 5, lines 78-82 – first with an example of an existing coordinated care pathway and then with an example of a lacking coordinated care pathway. On page 6, lines 121-123 the coordinated treatment is described in short again.

Reviewer comment #6

What are the potential advantages/ hypothesized benefits of using one-only exercise therapy specifically?

Author response

We thank the reviewer for this comment. We agree that it’s important to argue why only one exercise was used. The advantages of using a one exercise-only approach is that it’s simple and pragmatic. That is, it is easy to set up at home, easy to remember how to perform, requires little intellectual effort and is easy to master [38]. Many physical rehabilitation interventions comprise several exercises each with its own specific instruction, resulting in accumulation of time and intellectual effort needed to be able to complete the intervention effectively. Pertaining to this notion, Henry and colleagues found that older adults prescribed two exercises performed the exercises more accurately according to the exercise instruction than subjects prescribed eight exercises [39]. In line with this, it is expected that simple exercise prescriptions will facilitate a mastery of the exercise [38]. Furthermore, questioning the approach of multi-exercise interventions is whether there is an added muscular strength benefit of having several exercises that stresses the same muscle tissue as this is unknown and could be unnecessary [40]. Several exercises require more time dedicated to exercising and thus calls for a larger motivation. Furthermore, several exercises require a certain amount of surplus mental energy, which often is low in patients as their condition is demanding. Also, exercise can inflict pain also counteracting adherence. Thus, the rational for investigating a single knee-extension strength home-based exercise is that it could improve adherence as it is simple (minimal intellectual effort), does not take a long time (requires less surplus energy) and is likely to inflict less pain (less stress imposed on the knee joint). The knee-extensor exercise is chosen as the knee-extensor muscles are the single most important muscles related to knee pain and physical function in patients with knee OA [6,41–43].

We believe we have addressed this in the manuscript under the paragraph named “Context: The QUADX-1 Trial” on pages 6-7, lines 119-139. The arguments provided in the manuscript are less comprehensive than in the above paragraph as we refer to the protocol paper of the QUADX-1 Trial for further information on the single knee-extensor exercise [1]. 

Further, being physical therapists ourselves (some of the authors), it is our experience that the profession sometimes is too optimistic on behalf of the patients they treat. Afraid of not addressing a deficit that may or may not be linked to symptoms, we prescribe more exercises than justified by the scientific evidence - a better safe than sorry approach. Unfortunately, this approach may do no good for exercise adherence [39]. So, with the one-exercise approach, we wanted to challenge this, well aware that if the parent trial (QUADX-1) came out “negative” with regards to dosage-differences - but more importantly - without any effect on surgical status, the patients would say: You used one exercise only. What were you thinking? The QUADX-1 trial results are not published yet, but in confidence we can say that it works surprisingly well. 

Reviewer comment #7

Other:

Para 2, makes the argument of strong physical therapist-patient relationship in contrast to surgeon-patient relationship before moving on (par 3) to the need for coordination between the two. It is unclear what the purpose of such an argument is, since the intervention under scope is characterized by limited physical therapist- patient interaction.

Author response

We thank the reviewer for this comment. We agree that the purpose of this argument is unclear in the introduction of the manuscript. The argument related to the therapist-patient relationship is brought up again in the discussion – where we believe the purpose is clearer (Page 35-36, lines 766-785). 

Action taken

We have deleted the paragraph in the introduction (Page 5, lines 81-89).

Exercise therapy is generally provided by physical therapists and most often consists of different exercises (an exercise program) which can be supplemented with other treatment modalities, such as e.g. manual therapy [14,15,44]. This supervised and group-based organization of treatment enables physical therapists to interact and to engage themselves to a great extent in the treatment of their patients, creating a strong physical therapist-patient relationship [16]. This high level of engagement is possible because physical therapists spend a relatively long time with the patients during an exercise session [16]. This is in contrast to orthopedic surgeons who only have limited time with patients in their out-patient clinic to assess the need for surgical treatment [16,45]. 

Reviewer comment #8

The aim stated in line 106 (“identify perceived facilitators and barriers”) is different from aim stated in abstract (“investigate key stakeholder perspectives”). In the methods section this qualitative study is presented as aiming at “clinical applicability and implementation” (line 126). Please be clear and consistent about the study aims, which should also be reflected in the interview schedules, results and discussion.

Author response

We thank the reviewer for this comment. We agree that this is unclear. Please see our response to reviewer comment #1 for our corrections and alignment related to the aim stated in the abstract and manuscript. We also agree with the discrepancy related to the study aim in the method section. We have aligned this with the above corrections to the aim of the study.

Action taken

The paragraph “Context: The QUADX-1 trial” in the methods section was changed from:

The project is designed as an intervention trial with concurrent gathering of information for clinical applicability and implementation (the present qualitative study), also referred to as a hybrid I design [4].

To:

The project is designed as an intervention trial with concurrent identification of perceived facilitators and barriers (the present qualitative study), also referred to as a hybrid I design [4].

Changes made at page 7, lines 129-131.

Methods

As the aims of the study remain to be clarified, further revisions might be needed in the process of analysis. For example, if the study focuses on identifying barriers and facilitators to implementation the unit(s) of analysis might change, or a deductive approach could be adopted at some point in the process.

Author response

Based on comment from the associate editor and reviewer we have made major revision to the introduction and methods paragraphs and clarified the why we used an inductive approach.

As the aim of the analysis was exploratory, we applied an inductive approach [25]. If we had had a hypothesis which we wanted to test or had a framework or theory to base the analysis on we could have applied a deductive analysis [23,24].

Reviewer comment #9

That aside, the main issue here is the limited study sample, which raises questions about credibility of the findings, especially since data saturation is not discussed/ was not achieved. How was variability in participants’ experiences ensured? Some information on relevant participants’ characteristics (e.g. level of experience in their profession) and on hospital and rehabilitation centres (e.g., how were they selected? Considered typical?) could help in this direction. Further, the authors could consider (A) repeating the interviews with the participants post-intervention or (B) interview a wider sample of healthcare professionals outside the QUADX-1 trial, until saturation is achieved. The latter would be particularly useful in planning implementation of the proposed therapy in standard healthcare practice.

Author response

We thank the reviewer for this comment. We agree that this is important to highlight in the manuscript. We recognize the limitation in the sample size and the consequences it could have regarding data saturation. The participants were selected based on their ability to provide information about the topic under investigation. We sought situational representativeness rather than demographic representativeness meaning that participants were selected for their ability to provide information about the area under investigation [2]. As stated below we found this limitation necessary to investigate facilitators and barriers among the health care professionals involved in the clinical trial as these were the only possible participants to interview. We have made this potential limitation clearer in the manuscript. 

Information on the included municipalities and hospital as well as the timeframe have been added to the manuscript. Demographic details on the participants are now provided in Table 1. We have reported the manuscript using the SRSQ checklist, please see appendix 1.

Action taken

The highlighted text has been added to the “recruitment and study participants” paragraph. 

Thus, the six physical therapists and four orthopedic surgeons represents all possible participants as we sought situational representativeness rather than demographic representativeness (37). We recognize that a sample size of ten participants could be a limitation and inadequate to attain data saturation. However, we consider this necessary as this was the only possibility to investigate facilitators and barriers among the orthopedic surgeons and physical therapists involved in the QUADX-1 trial. When interpreting the results this should be kept in mind.

Changes made at page 8-9, lines 172-178.

The highlighted text has been added to the “recruitment and study participants” and “procedures” paragraphs. 

We recruited six physical therapists from three municipalities in the capital region of Denmark and four orthopedic surgeons from one university hospital in the capital region of Denmark involved in the QUADX-1 trial [1] (Table 1). Inclusion criteria: participants had to be involved in the QUADX-1 trial, as the intervention under study was not implemented in routine clinical practice. No exclusion criteria were applied. Thus, the six physical therapists and four orthopedic surgeons represents all possible participants as we sought situational representativeness rather than demographic [2]. We recognize that a sample size of ten participants could be a limitation and inadequate to attain data saturation. However, we consider this necessary as this was the only possibility to investigate facilitators and barriers among the orthopedic surgeons and physical therapists involved in the QUADX-1 trial. When interpreting the results this should be kept in mind. Participants were contacted by the primary investigator and interview moderator (RSH) by e-mail with an invitation to participate in the interviews between November 2016 and June 2017. RSH sent the invitations because he would be conducting the interviews. All invited participants accepted. All eligible physical therapists had daily clinical work with patients diagnosed with knee OA and rehabilitation following KR. All eligible orthopedic surgeons had daily clinical work with patients potentially eligible for KR due to knee OA symptoms. A random sample of patients participating in the QUADX-1 trial were also interviewed about their perceptions of facilitators and barriers towards coordinated non-surgical and surgical treatment using pre-operative home-based exercise therapy with one exercise. This work is as yet unpublished.

Changes made at page 8-9, lines 168-188

And

The interviews took place in meeting rooms at a university hospital in the capital region of Denmark.

Changes made at page 11, lines 221-222.

Reviewer comment #10

Also, a couple of discrepancies: the chosen method of data analysis is described at places as thematic analysis (e.g. lines 219, 234, 253, 264, 710), elsewhere as content analysis (abstract, citation 35). 

Author response

We thank the reviewer for this comment. We agree with the discrepancies and have changed the wording “content analysis” to “thematic analysis” in the abstract/manuscript. The reference related to the type of analysis has been changed from Graneheim et al. (content analysis) [46] to Nowell et al. (thematic analysis) [47].

Action taken

In the abstract the following changes have been made.

 Methods

This qualitative study is embedded within the QUADX-1 randomized trial that investigates a model of coordinated non-surgical and surgical treatment for patients eligible for KR. Physical therapists and orthopedic surgeons working with patients with knee osteoarthritis in their daily clinical work were interviewed (one focus group and four single interviews) to explore their perceived facilitators and barriers related to pre-operative home-based exercise therapy with one exercise-only in patients eligible for KR. Interviews were analyzed using thematic analysis.

Changes made at page 2, line 39.

Results

From the thematic analysis three main themes emerged: 1) Physical therapists’ dilemma with one home-based exercise, 2) Orthopedic surgeons’ dilemma with exercise, and 3) Coordinated non-surgical and surgical care. 

Changes made at page 2, line 41.

Reviewer comment #11

In the surgeons’ interview schedule (S3, Q4) self-management, including education, are mentioned. These are further reported in results and discussion. However, they are not described in the introduction and methods. Please ensure consistency and transparency in reporting.

Author response

We thank the reviewer for this clarifying comment. We agree with the comment and have added the below information to the manuscript to improve the consistency between the introduction and results/discussion sections. 

Action taken

The highlighted text has been added to the introduction paragraph. 

Both international and national guidelines recommend non-surgical treatment (i.e. exercise therapy, weight loss, self-management and education) before surgery is considered in patients eligible for KR [5–7,26–28] – and recent studies show that exercise therapy has been found to provide clinically relevant improvements in knee OA symptoms in patients eligible for KR [10–13,29,30].

Changes made at page 4, lines 63-67.

Other, minor comments:

Reviewer comment #12

Some details of the Context could fit better in the introduction, especially where literature review is involved (e.g., lines 131-133).

Author response

We thank the reviewer for this comment. We understand the point of moving the literature review part to the introduction. Despite this we suggest keeping the structure as it is. We believe it improves the reader friendliness to separate the more general introduction from information specifically related to the intervention under study (the QUADX-1 trial, the context). 

Reviewer comment #13

Line 220- was all text divided into meaning units?

Author response

We thank the reviewer for this comment. Parts of the text without coherent information or meaning were not divided into meaning units. This could be text parts with small talk and/or drift away from the topic under investigation. These were coded as “not for us”. As an example, during the focus group interview with the physical therapists this could be topical drift towards organization of treatment for patients with other diagnoses and complaining’s about general political decisions affecting their work (unrelated to the QUADX-1 trial). 

Reviewer comment #14

How many municipalities are served by/linked to this orthopaedic department (line 152)?

Author response

We thank the reviewer for this comment. The Copenhagen University Hospital Hvidovre receives patients from ten municipalities. The three collaborative municipalities in the QUADX-1 trial (Copenhagen, Hvidovre and Brøndby) were chosen as the demography in these three municipalities are representative for the population that the hospital service (e.g. socioeconomic status). 

Reviewer comment #15

Line 178 “experiences…on the coordinated… treatment”. Consider removing the term “experience”, since this is about preconceptions, not actual experiences with the trial (line 196).

Author response

We thank the reviewer for this comment. We agree that “experiences” is misleading since the purpose is about preconceptions (as the reviewer point out). We have removed the phrasing “experiences” from the sentence.

Action taken

The paragraph “Interviews: Focus group and single interviews” was changed from:

We aimed to use focus group interviews for all participants as the purpose of the interviews was to explore the perceived facilitators and barriers and associated feelings, experiences and attitudes of the health care professionals on the coordinated non-surgical and surgical treatment investigated in the QUADX-1 trial. 

To:

We aimed to use focus group interviews for all participants as the purpose of the interviews was to explore the perceived facilitators and barriers, associated feelings, opinions and attitudes of the health care professionals on the coordinated non-surgical and surgical treatment investigated in the QUADX-1 trial. 

Changes made at line page 10, lines 195-198.

Reviewer comment #16

Lines 229-232. More details on how these preconceptions changed, would be useful.

Author response

We thank the reviewer for this comment. We agree that more details on how the preconceptions changes increase the credibility of the process and findings. We have added an example of a change in preconceptions.

Action taken

The highlighted text has been added to the paragraph “Data analysis”.

Through this process, it was possible for RSH to put his preconception in dialogue with the text (fusion of horizons). Thus, the understanding of physical therapists and orthopedic surgeons perceived facilitators and barriers towards coordination of surgical and non-surgical treatment gradually changed [48]. As an example, one preconception was related to orthopedic surgeons’ view on exercise as not being useful in patients with severe knee OA. This preconception changed when the effect of exercise or lack hereof was mentioned as useful in the decision on surgery or not.

Changes made at page 12, lines 254-260.

Results

I acknowledge that data analysis was reviewed and agreed upon by more than one researcher. However, the reported results do not appear to clearly reflect (patterns found in) the data and the subthemes and themes appear to be poorly supported. Specifically:

Reviewer comment #17

Quotes do not represent all participants. There are no quotes from physical therapists 4 and 5. Instead quotes from selected participants appear repeatedly. For example, subthemes relevant to surgeons are supported by more than one quote from the same participant but not more than two participants are referenced (subthemes 3, 5, 6, 8, 9).

Author response

We thank the reviewer for this comment. We acknowledge that the representation of the quotes in the manuscript can be distinctive. The quotes showed in the manuscript are chosen as they are exemplary and thus representative for more participants than the one quoted. The quotes are chosen because they support central points in the analysis and therefore back a specific point. The participants in the focus group interview adopted different roles where participants 1, 2, 3 and 6 contributed more actively with answers to questions while participant 4 and 5 were more reticent. Participant 4 and 5 were more reticent with their answers despite numerous invitations from the moderator to contribute with their opinions. Participant 4 and 5 contributed more with statements acknowledging and supporting answers from the other participants. See below example:

Physical therapist 1: “I think it will be interesting to see when we can do with just one exercise”.

Physical therapist 4: “Well, I feel the same way”.

Reviewer comment #18

The subtheme title and content (i.e., authors description, original quotes) are often not in line and there doesn’t seem to be a connecting thread within each subtheme. In addition, there is overlap between suggested subthemes. Two examples: under the theme “professional role simplified” a substantial amount of text (lines 344-357) discusses physical therapists’ expressed preference towards more than one treatment options; the supporting quote is on therapists’ views regarding good candidates for the proposed treatment. The context of the next subtheme, “skepticism towards one home-based exercise”, includes again a discussion on physical therapists’ suggestions for more treatment options (lines 367-374), positive views of one exercise (376-94) and reference to “professional self-image” (376-78).

Author response

We thank the reviewer for this comment. We agree that the tread within some of the sub-themes was unclear. We have reviewed and revised the results section. We have renamed and re-arranged theme 1 and 2 and moved the associated sub-themes accordingly. 

We have changed the name of theme 1 from “Physical therapists’ and orthopedic surgeons’ ambivalence in their professional roles” to “Physical therapists’ dilemma with one home-based exercise”. The associated sub-themes are 1) Supporting patient self-management is a physical therapy core skill, 2) Professional role as a physical therapist is simplified, 3) Skepticism towards one home-based exercise and 4) Patient preferences. 

The name of theme 2 is changed from “Orthopedic surgeons view on exercise” to “Orthopedic surgeons’ dilemma with exercise”. The associated sub-themes are 1) Skepticism towards (long-term) effect of exercise in patients with severe knee OA, 2) Patient preferences and 3) Different purposes of referring a patient to exercise. 

Theme 3 and associated sub-themes are unchanged. 

Action taken

The revision and re-arrangement of the result section is comprehensive and thus we have refrained from copying the changes into this document. We refer to the uploaded file labelled “Revised Manuscript with Track Changes” where we believe the changes are easier to assess.

Reviewer comment #19

Parts of the results sections read more like discussion of the findings/ authors’ interpretations or review of the literature. Some examples: lines 277-280; 288-292; 452 (citation). On a similar note, the choice of wording in authors’ descriptions at points reads more like their own perspectives rather than participants’ perspectives. Some examples: 586-587 “exercise is used in a less constructive and inclusive way (by the surgeons)”; “professional dilemma” (612)- unclear what the dilemma is on, since surgeons’ quotes overall give their reasoning and clear-cut perspectives for using or not using a treatment (eg quotes on 549, 569, 579); “preferences” (line 605) rather than “clinical judgment”.

Author response

We thank the reviewer for this comment. We agree that parts of the result section read more like interpretation. We have removed the suggested sentences from the manuscript. Regarding the comment related to the “professional dilemma” presented in the summary: The dilemma emerges when the orthopedic surgeons follow guideline recommendations and this conflicts with their clinical expertise and patient preferences. We believe we have explained this at page 35, lines 766-769. 

Action taken

The following excerpts has been removed from the manuscript.

Exercise therapy is a central treatment modality in the physical therapy profession, not least in the treatment of patients with knee OA. Most exercise therapy is organized in group sessions where physical therapists monitor and adjust treatment closely, allowing a high degree of engagement in the treatment. However, in the interviews it turned out that the physical therapists recognize the importance of supporting patient self-management to complement supervised exercise therapy.

Changes made at page 16.

One sub-theme that emerged from the focus group interview was that the physical therapists are conscious about the importance of educating and providing patients with tools to self-manage their condition. In patients with chronic conditions (e.g. knee OA) self-management is especially important if the effect of treatment is to be sustained following supervised exercise. Exercise is likely a life-long treatment in patients with knee OA making provision of supervised exercise unrealistic, again underlining the importance of patient self-management. 

Changes made at page 16, lines 312-314.

Another view on exercise is that it is a treatment where the patient “can be parked” until surgery is needed. From this perspective, exercise is used in a less constructive and inclusive way, and more as a practical solution that can be used until the patient is ready for surgical treatment. 

Changes made at page 28, lines 604-605.

Reviewer comment #20

Original data provided is not in English, therefore not accessible to the majority of the readers.

Author response

We thank the reviewer for this comment. We acknowledge the limitation that the original data is not in English and thus not accessible to the majority of readers. The original data for all the interviews comprise >100 closely written pages and >48.000 words. The original data is made available without restrictions together with the manuscript providing readers with the possibility of translation. 

Reviewer comment #21

“Barriers” and “facilitators” are not reported in a consistent way in the results and the relevant research question is not answered comprehensively.

Author response

We thank the reviewer for this comment. We agree that in parts of the results section it was not clearly reported whether the results (sub-themes) were perceived as facilitators or barriers. We have made this clear and it is now reported consistently throughout the results section. That is, each sub-theme ends with sentence explaining whether it is a facilitator or a barrier. See below for changes made to the manuscript. 

Related to the comment that the research question is not answered comprehensively we have made the link between the study aim (perceived facilitators and barriers) and the results clearer. The first paragraph in the results section end with the following sentence: “The themes and sub-themes represent different perceived facilitators and barriers among orthopedic surgeons and physical therapists towards home-based pre-operative exercise in patients eligible for KR” (page 14, lines 292-294). Further, each paragraph related to theme 1-3 ends with a summary of the findings highlighting the perceived facilitators and barriers. Please see the summary paragraphs below. 

Action taken

Sub-theme 1 (page 17, lines 333-337)

As the physical therapists are aware of this, they embrace this skill and express that it is important to give patients a sense of responsibility for their own treatment and to teach them principles of self-management of their condition. In this way, even though they distance themselves from the patients, they keep some control over the patient’s treatment, and it becomes a potential facilitator. 

Sub-theme 2 (page 18, lines 366-367)

The predefined and advisory role with a limited number of consultations challenges and simplifies their professional role and, thus, becomes a potential barrier.

Sub-theme 3 (page 19, lines 387-389)

A home-based single exercise intervention without the option of exercise adjustment or the addition of other exercises becomes a potential barrier to physical therapists.

Sub-theme 4 (page 20, lines 419-423)

This underlines that according to the physical therapists not all patients are candidates for home-based exercise, which also supports the option of a stratified treatment approach (two treatment options). The possibility of two treatment options and thus a better chance of providing treatment suiting individual patient preferences becomes a potential facilitator.

Sub-theme 5 (page 24, lines 512-515)

Lack of belief in the effectiveness of exercise for patients with severe knee OA, doubt about the long-term effects of exercise and knowledge about the effectiveness of surgery create skepticism in the orthopedic surgeons, and these become potential barriers to referring patients with severe knee OA to exercise.

Sub-theme 6 (page 28, lines 587-590)

Thus, motivation is an important patient characteristic for the orthopedic surgeons, as they do not want to refer patients to a treatment that they are not going to adhere to due to lack of motivation. This would be a waste of everyone’s time and resources and becomes a potential barrier for referring patients to exercise. 

Sub-theme 7 (page 29, lines 616-618)

Assessing the effect of exercise on knee-related symptoms, evaluating patient resources, and providing patients with a “breathing space” to consider the treatment option of surgery all become potential facilitators for referring a patient to exercise.

Sub-theme 8 (page 31, lines 671-673)

Referring to exercise without knowing the content of the treatment provided and the time associated with referral to exercise becomes potential barriers for the orthopedic surgeons to refer patients to exercise.

Sub-theme 9 (page 34, lines 723-725)

Clear allocation of responsibility in relation to exercise referral and providing patients an optimized and transparent care pathway becomes potential facilitators for the coordinated non-surgical and surgical care pathway. 

Summary theme 1 (page 21, lines 425-434) 

In summary, a single exercise home-based intervention creates a dilemma among the physical therapists. On the one hand the physical therapists perceive the importance of providing patients with tools for self-management, the advantage of having two treatment options to meet patient preferences and the potential advantages of providing patients with only one exercise as facilitators for implementing the one exercise. These factors support the simplified treatment approach among the physical therapists and their view on their professional role. On the other hand, the physical therapists believe that the simplified treatment approach simplifies their professional role, limits contact time with patients and providing only one exercise limits use of professional skills. These barriers challenge the physical therapists creating ambivalence in their professional role.

Summary theme 2 (page 29-30, lines 627-639) 

In summary, results from our single interviews with the orthopedic surgeons show that adhering to clinical guideline recommendations - and at the same time using clinical expertise and considering patient preferences - creates a professional dilemma among the orthopedic surgeons. On the one hand, facilitators such as using exercise as a means to examine patient’s motivation for rehabilitation, providing patients with a low-risk-of-complications treatment while considering the option of surgery and knowledge of the effect of exercise can help guide the decision of surgery support the use of exercise as a treatment modality among orthopedic surgeons for patients with severe knee OA. On the other hand, barriers among the orthopedic surgeons towards referring patients with severe knee OA to exercise were skepticism towards the effect of exercise and especially the long-term effect in patients with severe knee OA and the dilemma of referring patients to exercise who are not motivated for this treatment modality. These barriers challenge the orthopedic surgeons creating ambivalence in their professional role. 

Summary theme 3 (page 34, lines 727-732) 

In summary, the orthopedic surgeons express frustration with variation in the treatment provided for the patients when they refer them to exercise in the municipality which becomes a barrier to referring patients to exercise. The physical therapists are positive in respect of the coordinated care pathway as they believe this will mean that patients are provided with quality care. This becomes a facilitator for coordinated non-surgical and surgical treatment. 

I would recommend that the authors:

Reviewer

Review the analytical process aiming to create more homogenous and comprehensive subthemes/ themes. Finding should reflect patters found across participants. This might not be that important if the aim is to identify trial implementation strategies, but this need to be clearer and methods modified accordingly

Author response

We believe we have addressed this in the responses to reviewer comment #18 and #19 and that the associated changes to the results section makes this more homogenous and consistent.

Reviewer

Provide a supplementary table where under each subtheme there is one quote from each participant that supports the subtheme

Author response

We believe we have addressed this in response to reviewer comment #17.

Reviewer

Provide a translation in English of a single interview and part of the focus group.

Author response

We believe we have addressed this in response to reviewer comment #20.

Reviewer

Be succinct when reporting the findings. Indicate which text represents views/ perspectives etc. expressed by the participants (e.g. participants described, agreed, emphasized/...). Also provide more indications of frequency/ homogeneity of the statements made (eg all/ most/ few physical therapists expressed…). Be more reflective and refrain from making interpretations in this section.

Author response

We believe we have addressed this in response to reviewer comment #18 and #19 along with the associated changes made in the manuscript.

Reviewer

Also, please keep a consistent and easy to follow structure. Examples of issues in the structure that are confusing for the reader: “physical therapists” section (line 276 onwards) and orthopedic surgeons (line 407 onwards) appear as subthemes, with their own introductory paragraph, although these are not previously mentioned- unclear what they represent; Subtheme numbering: in lines 254-260 the subthemes are numbered 1 to 9, Table 2 has no numbering, whereas in the remaining results sections subthemes are re-numbered from 1.

Author response

We believe we have addressed this in response to reviewer comment #18 and #19 along with the associated changes made in the manuscript.

Discussion

Reviewer

I believe this section will be revisited following revisions on the previous sections, therefore it won’t be of use commenting on the content as it is. Overall, a number of good points are discussed. Areas for refinement could be: clear take-home messages (3-4) that bring the discussion back to the points raised in the introduction (study aims, study population, existing gaps, concepts, trial implementation etc.).

Author response

We thank the reviewer for this comment. We agree that take-home messages are important for the readability of the manuscript. We believe we have addressed this in the summary at the end of the discussion paragraph (page 40-41, lines 883-899).

References

1. Husted RS, Troelsen A, Thorborg K, Rathleff MS, Husted H, Bandholm T. Efficacy of pre-operative quadriceps strength training on knee-extensor strength before and shortly following total knee arthroplasty: protocol for a randomized, dose-response trial (The QUADX-1 trial). Trials. 2018;19: 47. doi:10.1186/s13063-017-2366-9

2. Horsburgh D. Evaluation of qualitative research: Evaluation of qualitative research. Journal of Clinical Nursing. 2003;12: 307–312. doi:10.1046/j.1365-2702.2003.00683.x

3. WMA. World Medical Association. WMA Declaration of Helsinki - Ethical Principles for Medical Research Involving Human Subjects. 2017. Available: Available at: https://www.wma.net/policies-post/wma-declaration-of-helsinki-ethical-principles-for-medical-research-involving-human-subjects/. Accessed November 15, 2017

4. Curran GM, Bauer M, Mittman B, Pyne JM, Stetler C. Effectiveness-implementation Hybrid Designs: Combining Elements of Clinical Effectiveness and Implementation Research to Enhance Public Health Impact. Medical Care. 2012;50: 217–226. doi:10.1097/MLR.0b013e3182408812

5. Sundhedsstyrelsen - Knæartrose. Nationale kliniske retningslinjer og faglige visitationsretningslinjer. 1st ed. Kbh.; 2012. 

6. McAlindon TE, Bannuru RR, Sullivan MC, Arden NK, Berenbaum F, Bierma-Zeinstra SM, et al. OARSI guidelines for the non-surgical management of knee osteoarthritis. Osteoarthr Cartil. 2014;22: 363–388. doi:10.1016/j.joca.2014.01.003

7. Fransen M, McConnell S, Bell M. Exercise for osteoarthritis of the hip or knee. The Cochrane Database of Systematic Reviews. 2015;9: CD004376. doi:10.1002/14651858.CD004376.pub3

8. Carr AJ, Robertsson O, Graves S, Price AJ, Arden NK, Judge A, et al. Knee replacement. The Lancet. 2012;379: 1331–1340. 

9. Hawker G, Wright J, Coyte P, Paul J, Dittus R, Croxford R, et al. Health-Related Quality of Life after Knee Replacement: Results of the Knee Replacement Patient Outcomes Research Team Study*. The Journal of Bone and Joint Surgery (American Volume). 1998;80: 163–173. doi:10.2106/00004623-199802000-00003

10. Skou ST, Roos EM, Laursen MB, Rathleff MS, Arendt-Nielsen L, Simonsen O, et al. A Randomized, Controlled Trial of Total Knee Replacement. New England Journal of Medicine. 2015;373: 1597–1606. doi:10.1056/NEJMoa1505467

11. Skou ST, Roos EM, Laursen MB, Rathleff MS, Arendt-Nielsen L, Rasmussen S, et al. Total knee replacement and non-surgical treatment of knee osteoarthritis: 2-year outcome from two parallel randomized controlled trials. Osteoarthritis and Cartilage. 2018;26: 1170–1180. doi:10.1016/j.joca.2018.04.014

12. Dabare C, Le Marshall K, Leung A, Page CJ, Choong PF, Lim KK. Differences in presentation, progression and rates of arthroplasty between hip and knee osteoarthritis: Observations from an osteoarthritis cohort study-a clear role for conservative management. International Journal of Rheumatic Diseases. 2017;20: 1350–1360. doi:10.1111/1756-185X.13083

13. Gwynne-Jones JH, Wilson RA, Wong JMY, Abbott JH, Gwynne-Jones DP. The Outcomes of Nonoperative Management of Patients With Hip and Knee Osteoarthritis Triaged to a Physiotherapy-Led Clinic at Minimum 5-Year Follow-Up and Factors Associated With Progression to Surgery. The Journal of Arthroplasty. 2020 [cited 10 Mar 2020]. doi:10.1016/j.arth.2020.01.086

14. World Confederation for Physical Therapy W. Policy statement. Description of physical therapy. World Confederation for Physical Therapy; 2015. 

15. Higgs J, Refshauge, K, Ellis E. Portrait of the physiotherapy profession. Journal of Interprofessional Care. 2001;15: 79–89. doi:10.1080/13561820020022891

16. Miciak M, Mayan M, Brown C, Joyce AS, Gross DP. The necessary conditions of engagement for the therapeutic relationship in physiotherapy: an interpretive description study. Archives of Physiotherapy. 2018;8. doi:10.1186/s40945-018-0044-1

17. Cobos R, Latorre A, Aizpuru F, Guenaga JI, Sarasqueta C, Escobar A, et al. Variability of indication criteria in knee and hip replacement: an observational study. BMC Musculoskeletal Disorders. 2010;11. doi:10.1186/1471-2474-11-249

18. Hofstede SN, Marang-van de Mheen PJ, Vliet Vlieland TPM, van den Ende CHM, Nelissen RGHH, van Bodegom-Vos L. Barriers and Facilitators Associated with Non-Surgical Treatment Use for Osteoarthritis Patients in Orthopaedic Practice. Coles JA, editor. PLOS ONE. 2016;11: e0147406. doi:10.1371/journal.pone.0147406

19. Bunzli S, Nelson E, Scott A, French S, Choong P, Dowsey M. Barriers and facilitators to orthopaedic surgeons’ uptake of decision aids for total knee arthroplasty: a qualitative study. BMJ open. 2017;7: e018614. 

20. Littlewood C, Mawson S, May S, Walters S. Understanding the barriers and enablers to implementation of a self-managed exercise intervention: a qualitative study. Physiotherapy. 2015;101: 279–285. doi:10.1016/j.physio.2015.01.001

21. Slover J, Shue J, Koenig K. Shared Decision-making in Orthopaedic Surgery. Clinical Orthopaedics and Related Research®. 2012;470: 1046–1053. doi:10.1007/s11999-011-2156-8

22. Kitzinger J. Chapter 3. Focus groups. 3rd edition. Qualitative Research in Health Care. 3rd edition. Blackwell Publishing Ltd; 2006. 

23. Potter WJ. An Analysis of Thinking and Research About Qualitative Methods. Mahwah: Lawrence Erlbaum; 1996. 

24. Bryman A, Burgess RG. Analyzing Qualitative Data. London: Routledge; 1994. 

25. Gadamer H. Sandhed og metode, grundtræk af en filosofisk hermeneutik (Originaltitel: Wahrheit und Methode). Århus: Systime; 2004. 

26. Conaghan PG, Dickson J, Grant RL. Care and management of osteoarthritis in adults: summary of NICE guidance. BMJ. 2008;336: 502–503. doi:10.1136/bmj.39490.608009.AD

27. Ravaud P. Management of osteoarthritis (OA) with an unsupervised home based exercise programme and/or patient administered assessment tools. A cluster randomised controlled trial with a 2x2 factorial design. Annals of the Rheumatic Diseases. 2004;63: 703–708. doi:10.1136/ard.2003.009803

28. Hurley MV, Walsh NE, Mitchell H, Nicholas J, Patel A. Long-term outcomes and costs of an integrated rehabilitation program for chronic knee pain: A pragmatic, cluster randomized, controlled trial. Arthritis Care & Research. 2012;64: 238–247. doi:10.1002/acr.20642

29. Calatayud J, Casaña J, Ezzatvar Y, Jakobsen MD, Sundstrup E, Andersen LL. High-intensity preoperative training improves physical and functional recovery in the early post-operative periods after total knee arthroplasty: a randomized controlled trial. Knee Surgery, Sports Traumatology, Arthroscopy. 2016;25: 2864–2872. doi:10.1007/s00167-016-3985-5

30. Skoffer B, Maribo T, Mechlenburg I, Hansen PM, Søballe K, Dalgas U. Efficacy of Preoperative Progressive Resistance Training on Postoperative Outcomes in Patients Undergoing Total Knee Arthroplasty: Progressive Resistance Training Before TKA. Arthritis Care & Research. 2016;68: 1239–51. doi:10.1002/acr.22825

31. Altman R, Asch E, Bloch D, Bole G, Borenstein D, Brandt K, et al. Development of criteria for the classification and reporting of osteoarthritis: classification of osteoarthritis of the knee. Arthritis & Rheumatology. 1986;29: 1039–1049. doi:10.1002/art.1780290816

32. Dell’Isola A, Steultjens M. Classification of patients with knee osteoarthritis in clinical phenotypes: Data from the osteoarthritis initiative. Lammi MJ, editor. PLoS ONE. 2018;13: e0191045. doi:10.1371/journal.pone.0191045

33. Mandl LA. Determining who should be referred for total hip and knee replacements. Nature Reviews Rheumatology. 2013;9: 351–357. doi:10.1038/nrrheum.2013.27

34. Dieppe P. Who should have a joint replacement? A plea for more ‘phronesis.’ Osteoarthritis and Cartilage. 2011;19: 145–146. doi:10.1016/j.joca.2010.08.018

35. Gossec L, Paternotte S, Maillefert JF, Combescure C, Conaghan PG, Davis AM, et al. The role of pain and functional impairment in the decision to recommend total joint replacement in hip and knee osteoarthritis: an international cross-sectional study of 1909 patients. Report of the OARSI-OMERACT Task Force on total joint replacement. Osteoarthritis and Cartilage. 2011;19: 147–154. doi:10.1016/j.joca.2010.10.025

36. Mancuso CA, Ranawat CS, Esdaile JM, Johanson NA, Charlson ME. Indications for total hip and total knee arthroplasties. The Journal of Arthroplasty. 1996;11: 34–46. doi:10.1016/S0883-5403(96)80159-8

37. Ingelsrud LH, Roos EM, Gromov K, Jensen SS, Troelsen A. Patients report inferior quality of care for knee osteoarthritis prior to assessment for knee replacement surgery – a cross-sectional study of 517 patients in Denmark. Acta Orthopaedica. 2019; 1–6. doi:10.1080/17453674.2019.1680180

38. Newman S, Steed L, Mulligan K. Chronic physical illness: self- management and behavioural interventions. NewmanS,SteedL,MulliganK.Chronicphysicalillness:self- maMaidenhead: Open University Press; 2009. 

39. Henry KD, Rosemond C, Eckert LB. Effect of number of home exercises on compliance and performance in adults over 65 years of age. Physical Therapy. 1999;79: 270–277. 

40. Rhea MR, Alvar BA, Burkett LN, Ball SD. A Meta-analysis to Determine the Dose Response for Strength Development. Medicine & Science in Sports & Exercise. 2003;35: 456–464. doi:10.1249/01.MSS.0000053727.63505.D4

41. Culvenor AG, Ruhdorfer A, Juhl C, Eckstein F, Øiestad BE. Knee Extensor Strength and Risk of Structural, Symptomatic, and Functional Decline in Knee Osteoarthritis: A Systematic Review and Meta-Analysis: Risk of Deterioration in Knee OA and Knee Extensor Strength. Arthritis Care & Research. 2017;69: 649–658. doi:10.1002/acr.23005

42. Øiestad BE, Juhl CB, Eitzen I, Thorlund JB. Knee extensor muscle weakness is a risk factor for development of knee osteoarthritis. A systematic review and meta-analysis. Osteoarthritis and Cartilage. 2015;23: 171–177. doi:10.1016/j.joca.2014.10.008

43. Juhl C, Christensen R, Roos EM, Zhang W, Lund H. Impact of Exercise Type and Dose on Pain and Disability in Knee Osteoarthritis: A Systematic Review and Meta-Regression Analysis of Randomized Controlled Trials: Impact of Exercise Type and Dose in Knee Osteoarthritis. Arthritis & Rheumatology. 2014;66: 622–636. doi:10.1002/art.38290

44. PubMed.com Medical Subject Heading. Exercise Therapy (MeSH-term). 2019 [cited 29 May 2019]. Available: https://www.ncbi.nlm.nih.gov/mesh/?term=exercise+therapy

45. Klaber Moffett JA, Richardson PH. The influence of the physiotherapist-patient relationship on pain and disability. Physiotherapy Theory and Practice. 1997;13: 89–96. doi:10.3109/09593989709036451

46. Graneheim UH, Lundman B. Qualitative content analysis in nursing research: concepts, procedures and measures to achieve trustworthiness. Nurse Education Today. 2004;24: 105–112. doi:10.1016/j.nedt.2003.10.001

47. Nowell LS, Norris JM, White DE, Moules NJ. Thematic Analysis: Striving to Meet the Trustworthiness Criteria. International Journal of Qualitative Methods. 2017;16: 160940691773384. doi:10.1177/1609406917733847

48. Dahlager L, Fredslund F. Hermeneutisk analyse – og forståelse og for-forståelse. 4th ed. Forskningsmetoder i folkesundhedsvidenskab. 4th ed. Kbh.: Munksgaard Danmark; 2011.

---

## [Decision Letter · Decision Letter 1]

12 Oct 2020

Perceived facilitators and barriers among physical therapists and orthopedic surgeons to pre-operative home-based exercise with one exercise-only in patients eligible for knee replacement: A qualitative interview study nested in the QUADX-1 trial

PONE-D-20-02454R1

Dear Dr. Husted,

We’re pleased to inform you that your manuscript has been judged scientifically suitable for publication and will be formally accepted for publication once it meets all outstanding technical requirements.

Kind regards,

Andrew Soundy

Academic Editor

PLOS ONE

Additional Editor Comments (optional):

Reviewers' comments:

Reviewer's Responses to Questions

**Comments to the Author**

1. If the authors have adequately addressed your comments raised in a previous round of review and you feel that this manuscript is now acceptable for publication, you may indicate that here to bypass the “Comments to the Author” section, enter your conflict of interest statement in the “Confidential to Editor” section, and submit your "Accept" recommendation.

Reviewer #2: (No Response)

2. Is the manuscript technically sound, and do the data support the conclusions?

Reviewer #2: Partly

3. Has the statistical analysis been performed appropriately and rigorously? 

Reviewer #2: No

4. Have the authors made all data underlying the findings in their manuscript fully available?

Reviewer #2: Yes

5. Is the manuscript presented in an intelligible fashion and written in standard English?

Reviewer #2: No

6. Review Comments to the Author

Reviewer #2: Although there are no quantitative endpoints, there are serious design issues with this study. The investigators note that they recruited six physical therapists from three municipalities in the capital region of Denmark and four orthopedic surgeons from one university hospital in the capital region of Denmark involved in the QUADX-1 trial. What constitutes an adequate sample for this qualitative analysis and how is this number of participants justified ?

Also, the study is mostly descriptive, by intent. The long list of summary themes and conclusions is cumbersome to read. One would have a better feel for the reliability of the results if one could determine ‘how many said what’. For example how many physical therapists actually expressed skepticism towards over-simplified exercise therapy? Perhaps this can be determined from appendices 7 or 8. They are in Danish so this reviewer cannot be certain that such is the case.

7. PLOS authors have the option to publish the peer review history of their article (what does this mean?). If published, this will include your full peer review and any attached files.

Reviewer #2: No

---

## [Editor Report · Acceptance letter]

14 Oct 2020

PONE-D-20-02454R1 

Perceived facilitators and barriers among physical therapists and orthopedic surgeons to pre-operative home-based exercise with one exercise-only in patients eligible for knee replacement: A qualitative interview study nested in the QUADX-1 trial 

Dear Dr. Husted:

I'm pleased to inform you that your manuscript has been deemed suitable for publication in PLOS ONE. Congratulations! Your manuscript is now with our production department. 

Kind regards, 

on behalf of

Dr. Andrew Soundy 

Academic Editor

PLOS ONE